# Spatiotemporal history of fault-fluid interaction in the Hurricane fault, western USA

Jace M. Koger[1] and Dennis L. Newell[1]

[1]Department of Geosciences, Utah State University, Logan, UT 84322, USA

*Correspondence to*: Dennis L. Newell (dennis.newell@usu.edu)

**Abstract.** The Hurricane fault is a ~250-km-long, west-dipping, segmented normal fault zone located along the transition between the Colorado Plateau and Basin and Range tectonic provinces, western U.S. Extensive evidence of fault-fluid interaction, include calcite mineralization and veining. Calcite vein carbon ($\delta^{13}C_{VPDB}$) and oxygen ($\delta^{18}O_{VPDB}$) stable isotope ratios range from -4.5 to 3.8 ‰ and -22.1 to -1.1 ‰, respectively. Fluid inclusion microthermometry constrains paleofluid temperatures and salinities from 45–160 °C and 1.4–11.0 wt % as NaCl, respectively. These data suggest mixing between two primary fluid sources including infiltrating meteoric water (70 ± 10 °C ,~1.5 wt % NaCl, $\delta^{18}O_{VSMOW}$ ~-10 ‰) and sedimentary brine (100 ± 25 °C, ~11 wt % NaCl, $\delta^{18}O_{VSMOW}$ ~5 ‰). Interpreted carbon sources include crustal- or magmatic-derived $CO_2$, carbonate bedrock, and hydrocarbons. U-Th dates from 5 calcite vein samples indicates punctuated fluid-flow and fracture healing at 539 ± 10.8 (1-sigma), 287.9 ± 5.8, 86.2 ± 1.7, and 86.0 ± 0.2 ka in the upper 500 m of the crust. Collectively, data predominantly from the footwall damage zone imply that the Hurricane fault imparts a strong influence on regional flow of crustal fluids, and that the formation of veins in the shallow parts of the fault damage zone has important implications for the evolution of fault strength and permeability.

## 1 Introduction

Secondary mineralization, alteration products, and associated textures in fault rocks provide windows into past fault-fluid interaction in the crust. Fracture networks and associated sealing cements are widely recognized not only for their tectonic significance, but also for their impact on fluid movement and distribution in the crust of groundwater, hydrocarbons, and ore-deposits (Mozley and Goodwin, 1995; Benedicto et al., 2008; Caine and Minor, 2009; Eichhubl et al., 2009; Caine et al., 2010; Cao et al., 2010; Laubach et al., 2019). The rates, spatiotemporal evolution, and mineralogy of fracture sealing cements in fault zones control fault-zone strength, the buildup of pore-pressures, location and frequency of failure events, and the overall fault system architecture through time (e.g., Caine et al., 1996; Evans et al., 1997; Sibson, 2000). In order to constrain fault-fluid interaction during and after fault slip, we need to understand the sources of fluids moving through the systems, their temperature and chemistry, and the age of fracture in-filling minerals that aid in their healing. As highlighted in the next section, the microscopy, geochronology, stable and radiogenic isotope geochemistry, bulk-rock and micro-scale geochemistry, and fluid inclusion analysis of diagenetic products in fault zones collectively inform these processes.

Exhumed brittle faults and fault damage zones are excellent natural laboratories for interpreting the interaction between fluids and faults with implications for fault-zone permeability evolution, diagenesis, and the seismic cycle (e.g., Chester et al., 1993; Caine et al., 1996; Sibson, 1996; Caine et al., 2010; Mozafari et al., 2015; Salomon et al., 2020). Our research presented here is inspired by prior studies on exhumed normal faults in the western U.S., some of which are briefly highlighted below. Although we note that many excellent examples exist worldwide in settings such as the Apennines (e.g., Ghisetti et al., 2001; Smeraglia et al., 2018), Greenland (Salomon et al., 2020), and the Dead Sea (Nuriel et al., 2012). A common theme amongst these studies is the analyses of secondary carbonate cements and fracture filling veins. Carbonate mineralization is amenable to radiogenic and stable isotopic analyses, whole-rock elemental analysis, fluid inclusion work, and dating, which allows for interpretation of past fluid temperature, chemistry, sources, and timing of fluid flow in faults. For example, the Moab Fault located in the northeastern Colorado Plateau is a world-class natural analog for the interplay between hydrocarbon-bearing fluid movement, and permeability evolution along a fault zone (Foxford et al., 1998). This east dipping normal fault exhibits a protracted history of fluid-fault interaction including hydrocarbon residues, and carbonate, oxide, and siliceous diagenetic cements and veins associated with deformation features. A suite of prior studies interprets multiple episodes fluid migration and fault-rock diagenesis between the Permian and late Tertiary due to fluid expulsion from the Ancestral Rockies Paradox Basin, during Laramide deformation, and during post-Laramide extension and exhumation (Chan et al., 2000; Chan et al., 2001; Eichhubl et al., 2009; Bergman et al., 2013; Hodson et al., 2016).

Also located in the Colorado Plateau, the Little Grand Wash and Salt Wash faults are well-exposed examples of carbonate-cemented normal fault zones that are associated with modern day spring emanations and are instructive as natural analogs for geological carbon sequestration (Shipton et al., 2004). Here extensive carbonate veins, travertine spring mounds, and $CO_2$-rich springs and a $CO_2$ geyser (Crystal Geyser) are associated with normal faulting that taps a $CO_2$-rich fluid reservoir at depth. Fault slip, fracturing, and subsequent sealing via carbonate mineralization are interpreted to be linked to fluid pressure build-up and release. The cycle of fault slip and sealing is related to the rate of fracture filling (Frery et al., 2015), and may also be linked to changes in hydraulic head related to glacial-interglacial climatic fluctuations (Kampman et al., 2012).

To the east of the Colorado Plateau, along the Rio Grande rift, exhumed basin-bounding and intra-basin normal faults preserve a record of syntectonic changes to fault zone permeability due to groundwater flow and mineralization in poorly lithified siliciclastic sediments (Mozley and Goodwin, 1995; Heynekamp et al., 1999; Caine and Minor, 2009; Williams et al., 2015). These studies document progressive fluid-flow localized along faults due to deformation and carbonate cementation that result in compartmentalization of basin hosted aquifers. Recent work on the Loma Blanca fault, in the south-central Rio Grande rift, documents periodic fault-slip and calcite sealing using microscopy, isotope geochemistry, and U-Th geochronology (Williams et al., 2017a; Williams et al., 2017b; Williams et al., 2019). These studies suggest that deeply circulated, $CO_2$-rich fluids are focused up along this fault, and that the temporal record of calcite vein fills is linked to the earthquake cycle and fault-valve behavior in this part of the Rio Grande rift.

In this contribution, we present new results documenting paleofluid-fault interaction along the Hurricane fault zone, located at the transition between the Colorado Plateau and Basin and Range tectonic provinces of the western U.S. (Fig. 1). The Hurricane fault juxtaposes Mesozoic and Paleozoic carbonates, sandstones, and shales along its strike, with excellent exposures of the footwall. Also proximal to and offset by the Hurricane fault are Pliocene – Pleistocene

volcanic centers and basalt flows that may have periodically influenced the fluid-flow and thermal regime near the fault (Fig. 2). Prior research on the Hurricane fault has focused primarily on its structural and paleoseismic history (Stewart and Taylor, 1996; Fenton et al., 2001; Lund et al., 2007), with studies on fault-fluid interaction limited to modern thermal springs (Crossey et al., 2009; Nelson et al., 2009). We present the first quantitative results on the spatiotemporal thermochemical evolution of paleofluid flow and fluid-rock interaction along the Hurricane fault zone

using stable isotope geochemistry, fluid-inclusion microthermometry, and U-Th geochronology of calcite vein networks exposed in the footwall damage zone. Our data enables us to constrain the sources and approximately 540 ka evolution of fluid flow and fault-fluid interactions within the footwall of the Hurricane fault zone, and this study highlights the value of integrating relatively high-resolution U-Th dates with other geochemical data from fault-hosted calcite veins.

**2 Geological Setting of the Hurricane fault zone**

The Hurricane fault zone strikes roughly north-south in the transition zone between the Colorado Plateau and the Basin and Range tectonic provinces in southwest Utah and northwest Arizona (Fig. 1). Major tectonic events that have shaped the region include the Sevier orogeny, Laramide orogeny, and subsequent Basin and Range extension. The Sevier orogeny and associated fold and thrust belt initiated at ~125 Ma due to subduction and formation of a

continental arc along the western margin of North America (Armstrong, 1968; Heller et al., 1986). The fold and thrust belt progressed eastward until shallowing of the subducting Farallon slab marked the onset of the Laramide orogeny at ca. 75 Ma (Livaccari, 1991; Yonkee and Weil, 2015). Laramide deformation is marked by basement-cored uplifts and the formation of the Rocky Mountains. Hydration of the continental lithosphere during this time lab led to widespread magmatism following foundering of the Farallon slab (Humphreys et al., 2003). Basin and Range

extension and wide-spread normal faulting in the western U.S. began in the late Eocene (Axen et al., 1993).

Normal faults of the Basin and Range broadly follow Proterozoic accretionary and Sevier-Laramide compressional structural fabrics to accommodate late Paleogene extension (Armstrong, 1968; Quigley et al., 2002). Extension along the eastern margin of the Basin and Range adjacent to the Colorado Plateau initiated ~ 15 Ma (Axen et al., 1993). The

Colorado Plateau province has remained largely un-deformed by Basin and Range extension, and the transition from the thick, strong crust of the Colorado Plateau to the relatively thin crust of the Basin and Range occurs over a ~ 100-km-wide interval (Zandt et al., 1995). The eastern margin of the transition zone is also coincident with the Intermountain Seismic Belt, (Fig. 1), with multiple seismically active normal faults including the Wasatch and Hurricane fault zones (Smith et al., 1989). Late Cenozoic volcanism along the margin between the two tectonic

provinces is compositionally bimodal, indicative of high heat flow and partial melting of the mantle (Best and Brimhall, 1974).

The Hurricane fault is a 250-km long, segmented, west dipping normal fault in southwestern Utah and northwestern Arizona with poorly constrained origins in the mid-Miocene to Pliocene (Lund et al., 2007; Biek et al., 2010). Fault activity occurred predominantly in the Pleistocene, including up to 550 m of its total 600–850 m of throw (Lund et al., 2007). Six segments of the Hurricane fault are 30–40 km long and have been defined based on geometric and structural complexities at segmentation boundaries (Fig. S1) (Pearthree et al., 1983; Stewart and Taylor, 1996; Stenner and Pearthree, 1999). The Hurricane fault is recently active as evidenced by Quaternary scarps, and the magnitude ~5.8 earthquake occurring in 1992 east of St. George, Utah with a focus at ~15 km depth along the projected dip of the Hurricane fault surface (Stewart and Taylor, 1996). Long-term slip rates based on paleoseismic studies range from 0.44 to 0.57 mm/y (Lund et al., 2007).

Rock types juxtaposed by the fault include Paleozoic and Mesozoic sandstones, siltstones and mudstones, marine limestones, and evaporites (Biek, 2003; Biek et al., 2010). Exposures of hanging wall bedrock are broadly covered by Quaternary colluvium concealing Triassic units that are exposed in a few locations. Permian and Triassic units are well exposed in the footwall along the Hurricane cliffs, especially in canyons cutting the escarpment (Fig. S2). Units dominated by marine carbonates include the Permian Pakoon Dolomite, Permian Toroweap Formation, and Permian Kaibab Formation, and lower members of the Triassic Moenkopi Formation. Siliciclastic-dominated units in footwall exposures include the Permian Queantoweap Sandstone (stratigraphically equivalent to the Hermit Formation along the southern segments of the fault), and Triassic Moenkopi Formation. Where exposed along the fault, the Permian Queantoweap Sandstone and Hermit Formation are composed of very fine to fine-grained quartz-rich sandstone, locally cemented by calcite and/or silica and hematite.

Basaltic volcanism in the eastern Basin and Range in the transition zone to the Colorado Plateau began at ~15 Ma and has been most active within the last 2.5 My (Nelson and Tingey, 1997). Quaternary basaltic volcanic centers are spatially associated with the Hurricane fault (Fig. 2). Basalt flows are offset by the Hurricane faults, and these are used for constraining long-term slip rates (Lund et al., 2007). Volcanic eruptions are predominantly alkali-rich basalts with lesser basaltic andesite. Neodymium isotope ratios of Quaternary basalts reflect primarily lithosphere sources along the northern half of the Hurricane fault and asthenosphere/mixed source to the south (Crow et al., 2011). These periods of basaltic magmatism associated with Basin and Range extension may have created hydrothermal systems in the past that locally influenced groundwater chemistry and circulation in the Hurricane fault.

Prior work on fluid movement associated with this fault is limited to geochemical and isotopic studies of modern spring systems at Pah Tempe hot spring, near La Verkin, UT, and the Travertine Grotto and Pumpkin warm springs in Grand Canyon. At Pah Tempe hot spring, deeply-circulated meteoric waters emerge as $CO_2$-charged and saline fluids along the fault trace, and precipitation of calcite veins is evident in the exhumed fault rocks (Nelson et al., 2009). Travertine Grotto and Pumpkin warm springs are attributed to meteoric water mixing with deeply-sourced fluids that are flowing upwards along the basement-rooted Hurricane fault (Crossey et al., 2006; Crossey et al., 2009). Analyses

of volatiles exsolving from these springs identifies a predominantly deep (endogenic) source, with some modern contributions from mantle or magmatic sources.

## 3 Methods

### 3.1 Field Locations

Field investigations along the Hurricane fault were conducted between Cedar City, UT and the fault's intersection with Grand Canyon in Arizona (Fig. 2). Studies were restricted to well-exposed areas of the fault's footwall, typically where canyons and drainages cross the fault. Due to colluvial cover on the hanging wall, this study focused on footwall exposures of the fault and the footwall damage zone. Twenty-three field stations (Fig. 2) along Hurricane fault were investigated, and hand samples were chosen for subsequent microscopic and geochemical characterization of diagenetic alteration and secondary vein mineralization. Sampling criteria included vein morphology, cross-cutting vein relationships, varying vein/fracture orientations, and range of apparent diagenetic modification, including unaltered host rocks. The goal was to collect a representative suite of hand samples at each of the 23 field stations to capture the range of fault-fluid interaction observable over ~160 km of fault length. Sample locations were recorded using a Garmin[TM] GPS unit in decimal degrees relative to the WGS 1984 datum (Table S1). Fracture intensity (fractures per meter) in the footwall exposures were measured at each field station using linear scan line methods (e.g., Watkins et al., 2015). Fracture distance from fault and orientation was recorded for fractures intersecting a measuring tape oriented roughly orthogonal to the fault trace. Scan line lengths were variable due to available field exposures and ranged from <10 m to ~400 m.

### 3.2 Microscopy

Standard petrographic thin sections were made from 34 hand samples displaying a range of vein types and diagenetic alteration. Of these 34 samples, 15 doubly-polished thick sections (150-$\mu$m-thick) of calcite veins were prepared for fluid inclusion analyses. Thin section petrographic observations were made using Leica Z16 APO and Leica DM 2700P petrographic microscopes. Photomicrograph images were acquired with a Leica MC 170 HD camera and processed using the Leica Application Suite 4.6 software.

### 3.3 Fluid inclusion microthermometry

Fluid inclusions in secondary calcite mineralization were investigated using a Zeiss Universal transmitted light microscope with a Zeiss Epiplan 50x long-working distance objective. A USGS gas-flow heating and freezing stage was used to measure fluid inclusion homogenization and melting temperatures. The stage was calibrated to the critical point of water using a synthetic supercritical $H_2O$ inclusion (374.1 °C), the freezing point of a synthetic 25 mol % $CO_2$-$H_2O$ inclusion (-56.6 °C), and the freezing point of double-deionized water using an ice bath (0 °C). Using the 15 thick sections, 107 homogenization temperatures ($T_h$) and 35 melting temperatures ($T_m$) were determined from two-phase fluid inclusions in calcite (Table 1).

Fluid inclusion were classified and homogenization and melting temperatures were determined using criteria and procedures described by Goldstein and Reynolds (1994) and Goldstein (2001). After performing heating measurements, numerous 2-phase fluid inclusions with homogenization temperatures from 45 – 85 °C became metastable 1-phase liquid inclusions (i.e., the bubble did not re-nucleate upon cooling). In order to re-nucleate the second phase to facilitate measuring the melting temperatures, these fluid inclusions in these samples were intentionally stretched by heating to 110 °C for 18 hours in a laboratory oven (Goldstein and Reynolds, 1994). Stretching does not impact the melting temperature because it does not alter the inclusion composition. However, if the inclusion is damaged during stretching, resulting in some water leakage, this could render melting temperatures meaningless. For a few of these treated inclusions, unreliable melting temperatures >0 °C were obtained, and these were omitted from the data set. It is possible that stretching induced leakage from these inclusions, although this was not observed during petrographic observations. No pressure correction was performed to convert $T_h$ measurements to trapping temperatures ($T_t$). Assuming vein formation at a maximum depth of 800 m equivalent to the maximum throw on the fault (Anderson and Mehnert, 1976), a maximum pressure using a lithostatic load (2675 kg m$^{-3}$ rock density), and the maximum measured $T_h$ of 160 °C, the pressure correction is <10 °C (Fisher, 1976; Bakker, 2003) and considered insignificant for this study. $T_h$ measurements in this study are considered representative of $T_t$.

**3.4 Carbon and oxygen stable isotope analysis**

A Dremel® tool was used to collect 290 powdered sub-samples from calcite veins, mineralized fracture surfaces, limestone host rock, and calcite-cemented sandstone host rock. Carbon and oxygen stable isotope analyses of these samples was performed in the Utah State University Department of Geosciences Stable Isotope Laboratory using a Thermo Scientific Delta V Advantage Isotope Ratio Mass Spectrometer (IRMS) and a GasBench II using the carbonate-phosphoric acid digestion method (McCrea, 1950; Kim et al., 2015). Specifically, ~ 120-150 µg aliquots of relatively pure calcite samples and standards were placed into 12 ml Exetainer© vials and flushed with ultra-high-purity helium. Impure carbonate cements (e.g., calcite-cemented sandstone) required 300 to 8000 µg of sample to achieve acceptable peak amplitudes during analysis. After helium flushing, ~ 100 µL of anhydrous phosphoric acid was added to each sample and allowed to react for two hours at 50 °C before analysis. Carbon and oxygen stable isotope ratios were calibrated and normalized to Vienna PeeDee Belemnite (VPDB) using the NBS-19 and LSVEC, and NBS-19 and NBS-18 international standards, respectively (Kim et al., 2015). In house calcite standards were used to correct for drift and mass effects. Carbon and oxygen isotope ratios are reported using delta notation ($\delta^{13}C_{VPDB}$, $\delta^{18}O_{VPDB}$ values) in per mille (‰). Based on repeat analyses of in-house calcite standards, errors on $\delta^{13}C$ and $\delta^{18}O$ values are <0.1 ‰. For calculations of paleo-groundwater $\delta^{18}O$, which are reported relative to Vienna Standard Mean Ocean Water (VSMOW), calcite $\delta^{18}O_{VPDB}$ values are converted to the VSMOW scale using Eq. (1) (Sharp, 2007):

$$\delta^{18}O_{VSMOW} = 1.03091 * \delta^{18}O_{VPDB} + 30.91 \qquad \text{Eq. (1)}$$

**3.5 Uranium-thorium (U-Th) dating**

Pilot U-Th geochronology was conducted on 5 key calcite vein samples from two field locations (Table S2). These include locations 1-2 and 1-4, where veins are hosted in limestone and sandstone strata, respectively (Fig. 2). Veins

were slabbed with a rock saw and approximately 300 mg of calcite powder was collected from discrete veins or vein laminations using a Dremel® tool and submitted to the University of Utah ICP-MS laboratory for analyses. At location 1-2, one laminated vein was subsampled at two locations (one near vein wall and near the outer part of the vein) to

capture the timing of vein growth. At location 1-4, 3 generations of veins, determined based on cross-cutting relationships, were subsampled.

Chemical preparation and analyses were performed at the University of Utah following methods modified from Edwards et al. (1987) using a Thermo NEPTUNE Plus Multi-Collector-Inductively-Coupled-Mass-Spectrometer

(MC-ICP-MS). Powdered carbonate samples were dissolved in 16 M $HNO_3$ and equilibrated with a mixed $^{229}Th$-$^{233}U$-$^{236}U$ spike and refluxed on heat for at least one hour to ensure total dissolution. Uranium and thorium sample fractions were separated for analyses by anion exchange column chemistry. Measured peak heights were corrected for abundance sensitivities and mass bias, dark noise, background (blank) intensities, hydride contributions, ion-counter yields, and spike contamination. The spike was calibrated against solutions of CRM 145 and HU1 uraninite.

Uncorrected age uncertainties are reported as one standard error and include measurement error and uncertainties of activity. Details of the spike calibration and data treatment can be found in Quirk et al. (2020).

**4 Results**

**4.1 Fault zone diagenesis and veins**

Evidence for fluid-fault interaction along Hurricane fault zone exists at the macroscopic and microscopic scale.

Collectively referred to as "fault zone diagenesis" (Knipe, 1992) and "structural diagenesis" (Laubach et al., 2010), these observations form the foundation for subsequent geochemical and geochronological work. Examination of the fault zone exposures at the 23 field sites (Fig. 2) reveals that it is composed of a up to 400-m-wide damage zone. The damage zone (e.g., Caine et al., 1996) was qualitatively defined as the part of the exposed footwall away (east) from the main fault trace that exhibits minor slip surfaces, deformation bands, and is more intensely fractured than the

unaltered host rock. We acknowledge that our assessment of the damage zone thickness is limited to the observed and available fault exposures. Exposures of the fault core (e.g., Caine et al., 1996) are less common and range from 0.5 m to 2 m thick, characterized by fault breccia and gouge zones, often bounded by large slip surfaces. The record of paleofluid flow and deformation is best preserved in competent sandstone and limestone units within the damage zone, rather than finer grained units that are typically poorly exposed. Evidence for chemical and mechanical fluid-rock

interaction includes host rock alteration, veins, and mineralized/cemented slip surfaces, deformation bands, and breccias (e.g., Fig 3). Secondary minerals include calcite, with lesser hematite, manganese oxides, and gypsum. Reduction (e.g., "bleaching" of sandstones, Fig. 3 a, b) and oxidation features are observed in siltstone and sandstone strata with calcite and iron oxide cements. Manganese and iron-oxide vein cements, and brecciated veins are primarily observed in sandstone strata. Sparry calcite veins are the most common feature in limestone strata, with nearly every

fracture hosting some calcite mineralization. The calcite veins range from single generation mm- to cm-scale sparry fracture fills to cm-scale laminated and fibrous veins with clear crystal terminations. Vein walls comprise intact host

rocks (limestone and sandstone) and calcite-cemented breccia. Diagenetic products are most commonly associated with zones of more intense fracturing, although veins occur throughout the damage zone.

Fracture intensity varies from ~2 to 20 m$^{-1}$ within the Hurricane fault's damage zone. Intensely fractured zones (or corridors) of 10–20 m$^{-1}$ are 1–2 m wide and are pervasively mineralized and "bleached" if cutting hematite-cemented sandstone. Bleaching removes hematite quartz coatings, leaving white to tan coloration in contrast to the surrounding red sandstone. Fracture orientations typically follow two main sets: one striking 0 ± 10°, and one 300 ± 15°, both dipping-steeply 70 to 90° (Fig. S3). The approximately north striking fractures are generally similar to the map-scale

trend of the Hurricane fault. Please refer to the supplemental documentation for more descriptions and photos of the observed veins, features associated with fault slip, and alteration of the host rocks (Figs. S4 – S6).

**4.2 Vein geochemistry**

**4.2.1 Carbon and Oxygen stable isotope ratios**

Stable isotope ratios of carbon and oxygen were determined for calcite veins and host rocks from the field sites (see

data repository, Newell and Koger, 2020). The $\delta^{13}C_{VPDB}$ and $\delta^{18}O_{VPDB}$ values for the entire data set range from -4.5 to 3.8 ‰ and -22.1 to -1.1 ‰ ($\delta^{18}O_{VSMOW}$ = 8.1 to 29.8 ‰), respectively (Fig. 4 a). In the host rock units, carbonate cements in siliciclastic units and bulk limestone host-rock were analysed adjacent to veins and at ~ 1–2 m away for comparison. Host rocks near fractures have $\delta^{13}C_{VPDB}$ and $\delta^{18}O_{VPDB}$ values from -4.5 to 2.8 ‰ and -17.7 to -8.6 ‰, respectively. Away from fractures, host rock $\delta^{13}C_{VPDB}$ and $\delta^{18}O_{VPDB}$ values range from -2.0 to 3.8 ‰ and -8.5 to -1.1

‰, respectively. The $\delta^{13}C$ and $\delta^{18}O$ values for calcite veins, breccia cements, mineralized fractures, and slip surface cements span a wide range of values with considerable scatter. For the purposes of presentation and discussion these data are divided into 4 "vein sets" based on common lithological associations, vein morphologic features, and $\delta^{13}C$ and $\delta^{18}O$ value data patterns (Fig. 4 a; $\delta^{18}O$ values are presented vs. VSMOW to facilitate comparisons to paleofluid sources). Note that these 4 vein sets span multiple locations (Fig. S1) and show no correlation in C and O isotope

ratios with location. There is no apparent relationship between vein set and orientation (Fig. S3).

Vein set 1 calcite exhibits a positive correlation (slope = 1.6) between $\delta^{13}C_{VPDB}$ and $\delta^{18}O_{VSMOW}$ and is commonly intergrown with hematite when hosted in siliciclastic strata. Calcite in set 2 also displays a positive $\delta^{13}C_{VPDB}$ and $\delta^{18}O_{VSMOW}$ correlation (slope = 0.9) with $\delta^{13}C$ shifted to lower values compared to vein Set 1. Set 3 has a wide range

of isotopic values, showing no strong trends or patterns. Set 4 calcite $\delta^{18}O$ values that overlap with set 3 with $\delta^{13}C$ values that trend to significantly lower values. The majority of set 4 data are from location (1-2).

**4.2.2 Fluid inclusion microthermometry**

Of the 15 thick sections of calcite veins observed, 6 contain populations of two-phase fluid inclusions that yield homogenization ($T_h$) and melting ($T_m$) temperatures (Table 1, Fig. 5). Homogenization temperatures are used to

approximate the trapping temperature ($T_t$) and are representative of fluid temperatures during mineralization. Melting

temperatures depend on the nature and concentration of dissolved species and are used to estimate the salinity of paleofluids (Bodnar, 1993).

Fluid inclusion homogenization and melting temperature data is organized by the calcite vein set as described in section 4.2.1. Observed two-phase fluid inclusions range from ~5–40 μm on the long axis. Most inclusions are interpreted to be primary and there are few trails of secondary inclusions. Single-phase (water) fluid inclusions are also present with long axis dimensions <15 μm. Homogenization temperatures for set 1 two-phase inclusions range from 45–90 °C. Vein set 3 samples have two-phase fluid inclusion homogenization temperatures from 55–160 °C, and their distribution skews towards lower temperatures, with a mode at 65–70 °C. Only single-phase fluid inclusions are present in vein sets 2 and 4. Trapping temperatures for single-phase inclusions cannot be readily determined. Trapping temperatures are often inferred as <50 °C (Goldstein and Reynolds, 1994; Goldstein, 2001); however, small inclusions (<10 μm) can fail to nucleate a bubble up to ~140 °C (Krüger et al., 2007). Ice melting temperatures from vein set 1 range from -3–0 °C, equating to a salinity of 0 to 5 wt% as NaCl (Fig. 5). Calcite set 3 yield melting temperatures from -11 to 0 °C, equating to 0 to 15 wt% NaCl. Since no initial melting was observed, NaCl dominated salinity is assumed and calculated using the equation of Bodnar (1993).

**4.3 U-Th geochronology**

The U-Th dates from the 5 vein samples range from 86 ka to 539 ka (Table S2). More specifically, the laminated calcite vein from location 1-2, hosted in limestone strata, yields an inner lamination date of 113.1 ± 0.3 ka (1-sigma error) and an outer lamination date of 86.2 ±- 1.7 ka (Fig. 6). The three calcite veins at location 1-4, hosted in sandstone strata yield dates of 539 ± 10.8 ka, 287.9 ± 5.8 ka, and 86.0 ± 0.2 ka in chronological order consistent with cross-cutting relationships. Two dates from a single sample include calcite cement from a brecciated vein wall (288 ka) that is crosscut by a laminated calcite vein (86 ka) (Fig. 6). In the field, this vein crosscuts the 539 ka vein.

**5 Discussion**

**5.1 Paleofluid sources in the Hurricane fault**

The carbon and oxygen stable isotope ratios of the calcite veins can inform the groundwater composition, source, and processes at work during paleofluid circulation in the Hurricane fault. The C and O equilibrium isotopic fractionation between $CO_2$ and calcite (cc) and water and calcite (cc), respectively are temperature dependent, and assuming that isotopic equilibrium during mineralization is valid, additional information on the paleofluid temperature is needed to proceed. Homogenization temperatures of primary 2-phase fluid inclusions in calcite, when present, are a reliable method to estimate temperature, and thus to calculate the paleofluid O and C isotopic composition using Eq. (2) and Eq. (3):

O isotopes:     $1000ln\alpha_{H2O-cc} = 2.89 - \frac{2.78*10^6}{T^2}$  (O'Neil et al., 1969; O'Neil et al., 1975)          Eq. (2)

C isotopes:     $1000ln\alpha_{CO2-cc} = 3.63 - \frac{1.194*10^6}{T^2}$ (Deines et al., 1974)          Eq. (3)

where $\alpha_{x\text{-}y}$ is the temperature dependent fractionation factor between water and calcite ($H_2O$-cc), $CO_2$ and calcite ($CO_2$-cc), and T is temperature in Kelvin. For the fractionation factor magnitudes expected for these two systems, the difference in delta values between the phases (i.e., $\delta^{18}O_{H2O}$ - $\delta^{18}O_{cc}$ and $\delta^{13}C_{CO2}$ - $\delta^{13}C_{cc}$) is a good approximation for $1000ln\alpha$ (Sharp, 2007). For these calculations, $\delta^{18}O$ and $\delta^{13}C$ values are on the VSMOW and VPDB scales, respectively,

When fluid inclusion data are not available, temperatures may be estimated based on other constraints, such as estimates on mineralization depths and the geothermal gradient, but the resulting paleofluid isotopic estimates will be far more uncertain due to surface-ward advection of geotherms (Forster and Smith, 1989). Clumped carbonate isotopic methods ($\Delta_{47}$) can yield reliable temperature estimates from fault-zone calcite mineralization (Swanson et al., 2012; Hodson et al., 2016), but are not available for this study. In the absence of these constraints, a range of

temperatures or starting fluid isotopic compositions can be explored to provide some interpretations of the calcite stable isotope data, again resulting in considerable uncertainty.

For the 6 samples that hosted populations of two-phase fluid inclusions, microthermometry heating and freezing data are used to estimates fluid trapping temperatures and salinities of the paleofluids present in the Hurricane

fault. In combination with $\delta^{18}O_{VSMOW}$ values from the calcite hosting these fluid inclusions, the paleofluid $\delta^{18}O_{VSMOW}$ values are calculated using Eq. (2). Although calcite oxygen stable isotope measurements are conducted on micro-drilled aliquots, these are still bulk samples when considering the microscopic distribution of fluid inclusions. Also, in each sample the microthermometry results yield populations of fluid inclusions with some variation in homogenization temperature. Therefore, we cannot connect individual isotopic values to individual fluid inclusions.

Rather we use the mean and standard deviation of measured temperatures in each sample along with the calcite $\delta^{18}O_{VSMOW}$ value to estimate a range of paleofluid compositions (Table 1). Similarly, we associate this range of oxygen isotope values to the mean and standard deviation of the paleofluid salinity as estimated from fluid inclusion melting temperatures. The paleofluid $\delta^{18}O_{VSMOW}$ value and salinity (wt % as NaCl) estimates for these samples show a strong positive correlation ($R^2$ = 0.8; Fig. 7). We interpret this correlation as mixing between two endmember fluid types,

and that over the history of fluid-fault interaction represented by these calcite veins, different proportions of ~100 +/- 25 °C, saline (~11 wt % NaCl), high $\delta^{18}O_{VSMOW}$ (~5 ‰) fluids have mixed with 70 +/- 10 °C lower salinity (~1.5 wt% NaCl), lower $\delta^{18}O_{VSMOW}$ (~-10 ‰) ground waters.

We suggest that the endmember characterized by high $\delta^{18}O$ values and high salinity is consistent with sedimentary

formation water (brine) that originated from extensive meteoric water-rock interaction and oxygen isotope exchange with marine sedimentary sequences (e.g., Clayton et al., 1966). Assuming a 25 – 30 °C geothermal gradient and the range fluid inclusion temperatures, circulation depths for these ground waters ranges from 2 to 6 km. This geothermal gradient is consistent with most observations from geothermal exploration wells in the region (Sommer and Budding, 1994). This is adequate to infiltrate all of the Mesozoic and Paleozoic strata in the region, including

thick sections of marine carbonate and evaporite bearing units (Biek, 2003; Dutson, 2005; Biek et al., 2010). Infiltration into these marine units is a likely source for the salinity in these ground waters. The endmember characterized by relatively low-salinity and low $\delta^{18}O$ value is likely dominantly meteoric water. Using the same geothermal gradient, these ground waters have circulated to ~3.5 km based on fluid inclusion constraints. For comparison, Pah Tempe hot springs (Nelson et al., 2009), and Pumpkin and Travertine Grotto springs (Crossey et al., 2009) emanate along the Hurricane fault and have similar oxygen isotope composition and salinity to this endmember (Fig. 7). Based on comparisons of Pah Tempe hot spring $\delta^{18}O$ and $\delta^{2}H$ values with other local and regional meteoric waters, Nelson et al. (2009) interpret the source of the hot spring water as meteoric water that infiltrated during the last glacial interval. Based on geochemical geothermometry estimates, and the observed shift in hot spring water to higher $\delta^{18}O$ values, Nelson et al. (2009) suggest that Pah Tempe thermal waters circulation depths of 3-5 km with temperatures of 70-150 °C. This approach has also been employed at other faults to interpret paleofluid compositions. For example, coupled fluid inclusion microthermometry and stable isotope values from fault-hosted calcite along the Moab fault, UT, USA, point to a mixing process between upwelling basin brines with meteoric water (Eichhubl et al., 2009). Although we suggest that the saline fluids in the Hurricane fault are the product of water-rock interaction, it is important to note that an alternative interpretation is mixing between meteoric water and basin brine derived from the evaporation of paleo-seawater. The "high salinity" endmember we observe is consistent with basin brines, albeit on the low end of observed salinities (e.g., Bodnar et al., 2014). It is possible that we have not captured the true high salinity endmember in our sampling. To further evaluate the source of the saline fluids, additional data that is currently not available such as halogen content and isotopic composition (e.g., Cl/Br, $\delta^{37}Cl$ value) may be diagnostic (e.g., Yardley et al., 2000).

In terms of the carbon sources in these two fluids, there are alternative ways to interpret the relatively narrow range of calcite $\delta^{13}C_{VPDB}$ values (0.35 to 1.73 ‰). First, using the average calcite formation temperatures from fluid inclusions, we estimate the $\delta^{13}C_{VPDB}$ of dissolved $CO_2$ in the paleofluid from -6.1 to -4.3 ‰ using Eq. (3) (Table 1). However, unlike the oxygen isotope system that most likely reflects the water composition, carbon composition can be reflective of a carbonate host rock. For example, dissolved carbonate in equilibrium with limestone bedrock (i.e., strongly buffered by the host rock) will result in calcite veins with a $\delta^{13}C$ similar to the host limestone (e.g., Dietrich et al., 1983). In this case, calculating the carbon isotopic composition of and external $CO_2$ source may not be appropriate, and the vein value is simply representative of the source carbon. In this study, the host rock limestone $\delta^{13}C_{VPDB}$ values range from -2.7 to 3.8 ‰ with an average of 1.2 ‰, which is in the range of expected values from marine carbonates (e.g., Hoefs, 1987; Sharp, 2007). However, in parts of the fault that have higher water-rock ratios or are generally carbonate poor (e.g., siliciclastic host rock), the carbon isotopes of calcite veins can be representative of an external $CO_2$ source dissolved and traveling in the groundwater. With these uncertainties in mind, we interpret the endmember carbon sources for the calcite veins as external $CO_2$ sources and local marine limestones. Based on the results from this study, there may be a weak association between the two carbon sources and the fluid

endmembers based on oxygen isotopes and salinity. In some but not all cases, vein $\delta^{13}$C values that are similar to host limestone tend to be associated with the highest salinity fluids. Veins hosted in sandstone units and associated with an external $CO_2$ source (~-6 ‰) are in most cases associated with the lower salinity fluids. These carbon isotope values are similar to the observed $\delta^{13}C_{VPDB}$ values of $CO_2$ at Pah Tempe (-5.5 ‰) and Pumpkin (-6.1 ‰) springs (Crossey et al., 2009; Nelson et al., 2009). These values overlap with mantle $CO_2$ values (Marty and Jambon, 1987),

but are also are similar to values observed in many crustal fluids and continental hot springs globally (Sherwood Lollar et al., 1997; Ballentine et al., 2002; Newell et al., 2008; Newell et al., 2015). Based on helium and carbon isotopes, Crossey et al. (2009) and Nelson et al. (2009) suggest that mantle $CO_2$ could range from a just a few percent to as high as ~40 % in the Hurricane fault hosted hot springs, depending on the mantle and crustal end members used. We do not have constraints on the helium isotope ratios of the paleofluids, so we cannot further evaluate the

possibility of magmatic contributions.

**5.2 Subsurface processes impacting isotopic values**

As shown earlier, the $\delta^{13}$C and $\delta^{18}$O values from calcite veins and cements associated with the Hurricane fault display a large range of values (Fig. 4 a). In addition to the binary mixing described in section 5.1, precipitation of calcite from fluids with a range of temperatures is occurring along flow paths. A fairly wide range of temperatures is evident

from the fluid inclusion work on vein sets 1 and 3. For a given water $\delta^{18}O_{VSMOW}$ and $\delta^{13}$C (of dissolved $CO_2$), varying temperature in Eqs. (2) and (3) results in trends in calcite $\delta^{18}O_{VSMOW}$ and $\delta^{13}C_{VPDB}$ with a slope of ~2.3 (Fig 4 b). To explore the impacts of both temperature change and mixing, the calcite forming from the saline ($\delta^{18}O_{VSMOW}$ = 5 ‰) and meteoric water ($\delta^{18}O_{VSMOW}$ = -10 ‰) endmembers, both with a $\delta^{13}C_{VPDB}$ = -6 ‰ are superimposed on the observed data (Fig. 4 b, shaded region). The vein set 1 pattern is fairly well matched by calcite forming over a the

range of T consistent with the fluid inclusion measurements (90-45 °C) from the low-salinity meteoric water end member ($\delta^{18}O_{VSMOW}$ = -10 ‰ ; $\delta^{13}C_{VPDB}$ = -6 ‰). The scattered values observed for vein set 3 are encompassed by the calcite forming from mixed saline and meteoric fluids over the range of temperatures consistent with the fluid inclusion measurements (160-50 °C). Although not shown on the diagram, using a limestone-buffered $\delta^{13}C_{VPDB}$ (~1 ‰) predicts values that are *not* consistent with any of the observed data, and this suggests that an external source

of $CO_2$ may be most appropriate for veins in both the carbonate and siliciclastic host rock. The vein set 4 isotopic values are best explained by mixing the saline end member at ~100 °C with a $\delta^{13}C_{VPDB}$ source of ~ -12 ‰ (Fig. 4 b). This low $\delta^{13}$C value is consistent with derivation from organic matter (Boles et al., 2004). Hydrocarbons are present regionally and in the strata that hosts the Hurricane fault (Bahr, 1963; Blakey, 1979). Mobilization and microbial oxidation of these hydrocarbons to form dissolved carbonate (Baedecker et al., 1993; Tuccillo et al., 1999) has been

shown in other fault settings to form calcite veins with low $\delta^{13}$C values (e.g., Eichhubl et al., 2009).

In addition to fluid mixing over a range of temperatures, other processes that occur during vein formation can also result in a range of calcite O and C stable isotope values. For example, open system (Rayleigh) $CO_2$ degassing and

calcite precipitation results in progressive fractionation of C and O stable isotopes in the fluid that result in

correlations between $\delta^{13}C$ and $\delta^{18}O$ values (Hendy, 1971). Kampman et al. (2012) used a Rayleigh fractionation model, assuming isotopic equilibrium, to describe C and O stable isotope values observed in fault-controlled aragonite veins and travertine deposits in the Salt Wash Graben, UT, USA. In this system, coupled $CO_2$ degassing and carbonate precipitation from a homogenous $CO_2$-charged fluid source can explain the positive correlation and range in $\delta^{13}C$ and $\delta^{18}O$ values. In combination with U-Th geochronological constraints, these authors suggest that this

system has been active periodically for >100 ky with a consistent paleofluid source and isotopic composition.

We test if the positively correlated C and O values observed for vein set 1 (slope = 1.6) and set 2 (slope = 0.9) (Fig 4 a) can be explained by similar processes. Rayleigh fractionation trends are included on figure 4 (b) for calcite resulting from $CO_2$ degassing (CO2-DIC), coupled $CO_2$ degassing and calcite precipitation (CO2-DIC-CC), and calcite

precipitation from groundwater (DIC-CC). These are calculated for both the carbon and oxygen isotope system using a Rayleigh distillation approach similar to Kampman et al. (2012). For example, for the progressive formation of calcite from bicarbonate, the carbon isotope ratios can be calculated from Eq. (4):

$$\delta^{13}C = \delta^{13}C_o - [1000 \times ln\alpha_{product-HCO3}(1-F)]$$      Eq. (4)


where $\delta^{13}C_o$ is the starting carbon isotope ratio for $HCO_3^-$ in solution, and F is the fraction of the C remaining in solution. Temperature dependent equilibrium fractionation factors ($\alpha$) for carbon isotopes in the calcite, $CO_2$, dissolved inorganic carbon (DIC) system are derived from Deines et al. (1974). We assume that the paleofluids were slightly acidic (~6.5) and near the equivalence point of $H_2CO_3$ and $HCO_3^-$ to compute the $\alpha_{DIC-CO2}$ (Gilfillan et al., 2009).

This is consistent with pH and geochemical observations at Pah Tempe hot springs and Pumpkin spring (Crossey et al., 2009; Nelson et al., 2009). Assuming the precipitation of calcite based on: $CaCO_{3(calcite)} + CO_{2(g)} + H_2O \Leftrightarrow 2HCO_3^-_{(aq)} + Ca^{2+}_{(aq)}$, the formation of calcite from $HCO_3^-$ in solution partitions, on a molar basis, the carbon equally between calcite and $CO_2$. Using the approach of Kampman et al. (2012), the net fractionation factor between the products and the bicarbonate in solution based on Eq. (5) is:

$\alpha_{product-HCO3}$ = 1/2 ($\alpha_{calcite-HCO3-}$) + 1/2($\alpha_{DIC-CO2}$)      Eq. (5)

Similarly, the oxygen isotope fractionation factor is calculated using Eq. (6):

$\alpha_{product-HCO3}$ = 1/3($\alpha_{CO2-HCO3}$) + 1/2($\alpha_{calcite-HCO3}$) + 1/6($\alpha_{H2O-HCO3}$)      Eq. (6)

The oxygen isotope system temperature dependent fractionation factors are derived from published relationships (O'Neil et al., 1969; O'Neil et al., 1975; Beck et al., 2005).

Focusing on vein set 1 and a range of temperatures observed from fluid inclusion, the computed trends alone cannot explain the observed data (Fig. 4 b). Rayleigh fractionation trends under equilibrium conditions for $CO_2$ degassing and coupled $CO_2$ degassing + calcite precipitation <85 °C yield positive correlations that are *not* similar to the observed slope or the wide range of values in set 1 calcite veins. Progressive calcite precipitation without CO2 degassing produces *negative* correlations in $\delta^{13}C$ and $\delta^{18}O$. Based on this analysis, it is unlikely that these processes are primary phenomena involved in the formation of calcite veins in the Hurricane fault zone. However, it is reasonable to assume that these processes are operable and impart a secondary control on the isotopic values observed from the fluid mixtures and may assist in explaining some of the data variability. It is important to note that these Rayleigh fractionation models assume isotopic equilibrium. Rapid degassing and calcite precipitation may result in disequilibrium and kinetic fractionation that cannot be quantitatively addressed, and that could produce a wider range of isotopic values. However, rapid degassing likely would produce populations of vapor-dominated fluid inclusion along with liquid-dominated inclusions in the calcite veins, which is not observed.

To summarize these analyses and interpretations, most of the vein C and O isotopic compositions and trends observed can be explained by a combination of the mixing of two primary fluid endmembers over a range of temperatures, with second order impacts from processes such as degassing during calcite precipitation. It is important to note that all of these processes would occur under open-system conditions in the fault zone. Vein set 1 is best explained by formation over a range of temperatures from the low-salinity endmember. Most of the values in set 2 and 3 can be explained by a mixture of the low-salinity meteoric and the sedimentary brine end members and precipitation over a range of temperatures. Vein set 4 requires addition of a much lower $\delta^{13}C$ component to the fluids responsible for vein set 3, likely derived from an organic source.

**5.3 Implications of vein geochronology**

Pilot U-Th geochronology on 5 samples indicates that calcite veins formed from 539 to 86 ka. These samples are from two different sample locations (1-2 and 1-4) separated by 13 km along strike (Fig. 2, 6), and from vein sets 1, 3, and 4. Specifically, at location 1-2, hosted in limestone, calcite vein growth occurred at 133 ka and 86 ka (set 4). Based on the interpreted growth direction of this vein, the dated laminations are chronologically consistent and suggest that the numerous vein laminations formed over a period of ~47 ka (Fig. 5 b). At location 1-4, hosted in sandstone, veins formed at 539 and 288 ka (set 1) and 86 ka (set 3). As described in the results, the dates are consistent with interpreted observed cross-cutting relationships, including the 86 ka calcite lamination cutting 288 ka calcite cemented brecciated sandstone (Fig. 6 a).

Based on the stable isotope results and analyses, the 539 and 288 ka veins are likely associated with the low-salinity meteoric water endmember (Fig. S7) and formed at moderate temperatures (60-70 °C). The 113 ka and both 86 ka veins are best associated with ~100 °C saline groundwater, with varying contributions of a low $\delta^{13}C$ carbon source

(Fig. S7). The 86 ka sample from location 1-2 has the lowest $\delta^{13}C$ (-7 ‰) observed in this study, and as discussed in section 5.2 requires an organic carbon source not observed at other locations.

Although this data set is small, it suggests punctuated vein forming events with some consistency along fault strike. Interestingly, these two fault locations appear to preserve the 86 ka fluid flow event, within analytical error, both requiring similar composition and temperature fluids, suggesting that the fluid circulation events have continuity over at least ~13 km of fault zone strike. More geochronological work is needed to evaluate if these crosscutting relationships are more broadly consistent along the fault zone. These dates can be used along with constraints on fault
slip rate to estimate the maximum depth of vein formation. Using the published slip rate estimates of 0.44 to 0.57 mm/y (Lund et al., 2007), this equates to vein formation depths of ~40 to 300 m. However, this assumes negligible exhumation of the footwall over the last 540 ka, and are minimum estimates. Using the local incision rate of the Virgin River through the footwall of 338 m/Myr (Walk et al., 2019) as a maximum estimate of footwall exhumation, this suggests a maximum depth of vein formation of 70 to 480 m. Consistent with the findings at thermal springs along
the Hurricane fault-zone (Crossey et al., 2006; Crossey et al., 2009; Nelson et al., 2009), this indicates that deeply-circulated thermal fluids are moving up the fault zone, advecting deeper geotherms towards the surface, and mixing with shallow meteoric fluids. The relatively shallow depth of these processes is notable, and has been observed at other major normal faults. For example, Smeraglia et al. (2018) documented Pleistocene synkinematic calcite mineralization along the Val Roveto fault (Apennines, Italy) that formed within the upper 350 m of the fault zone
during mixing of deeply-derived fluids and meteoric infiltration.

Although not the primary objective of this paper, the calcite vein textures in context of the preliminary geochronological results warrant a brief discussion. The vein wall breccias and laminated calcite veins observed along the Hurricane fault share similar characteristics to those in other major fault zones that have been attributed to co-
seismic or post-seismic sealing (e.g., Nuriel et al., 2011; Nuriel et al., 2012; Smeraglia et al., 2018). These fracture openings filled with laminated growth bands of fibrous calcite crystal are indicative of post-fracture opening sealing (crack-seal cycle) (Ramsay, 1980). These suggest fluid-pressurization and fluid flow cycles associated with periodic fracturing in the fault damage zone, possibly due to seismic activity (e.g., Sibson, 1994). Williams et al. (2017b) showed that detailed U-Th dating of these types of laminated veins inform the periodic nature of fracture opening and
sealing via calcite precipitation and argue these are associated with seismic events. Specifically, they documented 13 seismic events between 550 ka and 150 ka and use this to estimate long-term earthquake recurrence intervals. In addition to earthquakes, Pleistocene climatic cycles could influence groundwater flow and fluid pressure, and possibly be associated with vein forming events similar to what is observed at the Little Grand Wash and Salt Wash faults (e.g., Kampman et al., 2012). We have documented 4 such events in the last ~540 ka, suggesting that a similar high-
resolution geochronological study could yield meaningful information about the long-term recurrence of fluid-flow triggering events along the Hurricane fault zone, whether triggered by seismicity or linked to climatic cycles.

## 6 Summary and Conclusions

Integrated calcite vein stable isotope geochemistry, fluid inclusion microthermometry and U-Th geochronology document the nature of paleofluids circulating in the Hurricane fault over the last ~540 ky. Our results indicate that calcite veins form in the footwall damage zone of the fault from mixtures of two main fluids over a range of temperatures, and that processes such as $CO_2$ degassing may influence vein formation from the fluid mixtures. These include a relatively low salinity meteoric-affinity groundwater and a salty sedimentary formation water. Carbon sources are more ambiguous, but likely include significant contributions from crustal or magmatic $CO_2$ and carbonate bedrock, along with lesser amounts from hydrocarbons. Fluid inclusion microthermometry temperatures from ~45 to 160 °C indicating that these fluids have circulated deeply (up to 6 km) prior to flowing up the Hurricane fault zone. Our pilot geochronology is sparse (5 dates) but supports punctuated vein forming events at 539, 288, 113, and 86 ka. Considering the published long-term slip rates along the Hurricane fault, these veins likely formed in the upper ~500 m of the crust. Present-day up flow of similar composition fluids occurs at Pah Tempe hot spring and where the fault cuts the Colorado River in Grand Canyon at Pumpkin and Travertine Grotto springs.

These results have implications for how the paleohydrology of the Hurricane fault changes spatially and through time. Calcite cemented fault breccia and laminated, fibrous calcite veins are suggestive of cycles of fracture opening and healing (i.e., crack-seal textures). Deep groundwater circulation and fault processes result in high pore pressures in the fault zone, and subsequent fracturing triggers up flow of $CO_2$-charged thermal fluids, fluid-rock interaction in the fault zone, and mixing with other ground waters. Calcite mineralization and veining from these flowing fluids heals breccias and fractures. The multiple generations of cross-cutting veins and laminated veins indicates that healed parts of the fault have experienced this cycle multiple times, and that these processes have strongly impacted the flow properties of the fault zone. Data from this study show that these linked mechanical and hydrological processes are occurring in the upper ~500 meters of the fault zone and is occurring periodically over ~180 km of fault strike. We conclude that the Hurricane fault imparts a strong influence on regional flow of crustal fluids, and that the formation of veins in the shallow parts of the fault damage zone has important implications for fault strength in the upper most part of the crust.

## 7 Data Availability

Readers are invited to access the full data set archived on the EarthChem Library: https://doi.org/10.26022/IEDA/111542. (Newell and Koger, 2020).

## 8 Supplemental Information

Supplemental information, tables, figures are available at the following link: XXXXXX.

**9 Author Contributions**

Jace Koger conducted the field sample collection, sample preparation, and sample analyses as part of the requirement for his MSc in Geology from Utah State University (Koger, 2017). Dennis Newell was the thesis supervisor, provided assistance and mentorship on sampling and analytical techniques, guidance on data analysis, and is the corresponding author for the preparation of this manuscript.

**10 Competing Interests**

None.

**11 Acknowledgements**

We thank Andrew Lonero (USU Geosciences) and Diego Fernandez (University of Utah) for their assistance with stable isotopic and U-Th analyses, respectively. Funding for this research was provided by a Geological Society of America Student Research Grant to J. Koger and the Department of Geosciences at USU. This manuscript greatly benefitted from constructive reviews by Billy Andrews, Matthew Steele-MacInnis, and the topical editor Peter Eichhubl.

**Tables and Figures**

**Table 1. Calculated paleofluid $\delta^{13}$C and $\delta^{18}$O from calcite C and O stable isotopes and microthermometry**

| Field Station | Sample ID | $\delta^{13}C_{cc}$ ‰ (VPDB) | $\delta^{18}O_{cc}$ ‰ (VSMOW) | $T_h$ (°C) | $T_m$ (°C) | wt % NaCl[a] | $\delta^{13}C_{CO2}$[b] ‰ (VPDB) | $\delta^{18}O_{H2O}$[c] ‰ (VSMOW) |
|---|---|---|---|---|---|---|---|---|
| 1-2 | JK15HR41 | 0.55 | 22.6 | 102.5 ± 25.0 | -7.5 ± 1.7 | 11.0 ± 1.4 | -4.3 ± 1.1 | 5.6 ± 2.7 |
| 1-4 | JK15HR110 | 0.50 | 10.0 | 75.8 ± 2.3 | -2.2 ± 0.9 | 3.7 ± 0.5 | -5.7 ± 0.1 | -10.0 ± 0.3 |
| 3-1 | JK15HR151 | 1.27 | 9.8 | 66.7 ± 11.0 | -0.8 ± 0.3 | 1.4 ± 0.6 | -5.4 ± 0.7 | -11.4 ± 1.5 |
| 3-4 | JK15HR160 | 0.35 | 20.2 | 71.6 ± 9.8 | -3.2 ± 1.2 | 5.2 ± 1.8 | -6.1 ± 0.6 | -0.5 ± 1.3 |
| 3-5 | JK15HR169 | 0.77 | 18.3 | 72.2 ± 8.7 | -2.0 ± 1.9 | 4.7 ± 2.6 | -5.6 ± 0.5 | -2.3 ± 1.2 |
| 5-2 | JK15HR255 | 1.73 | 13.5 | 70.1 ± 7.0 | -1.0 ± 0.2 | 1.7 ± 0.7 | -4.8 ± 0.4 | -7.2 ± 1.0 |
| - | Pah Tempe HS[d] | - | - | - | - | 0.8 | -5.5 | -13.0 |
| - | Pumpkin Spr[e] | - | - | - | - | 1.1 | -6.1 | -10.6 |
| | Travertine Grotto[e] | - | - | - | - | 0.2 | - | -10.8 |

[a,b,c] calculated using Eq. (2), (4), and (3), respectively
[d]Nelson et al. (2004); [e]Crossey et al. (2009)


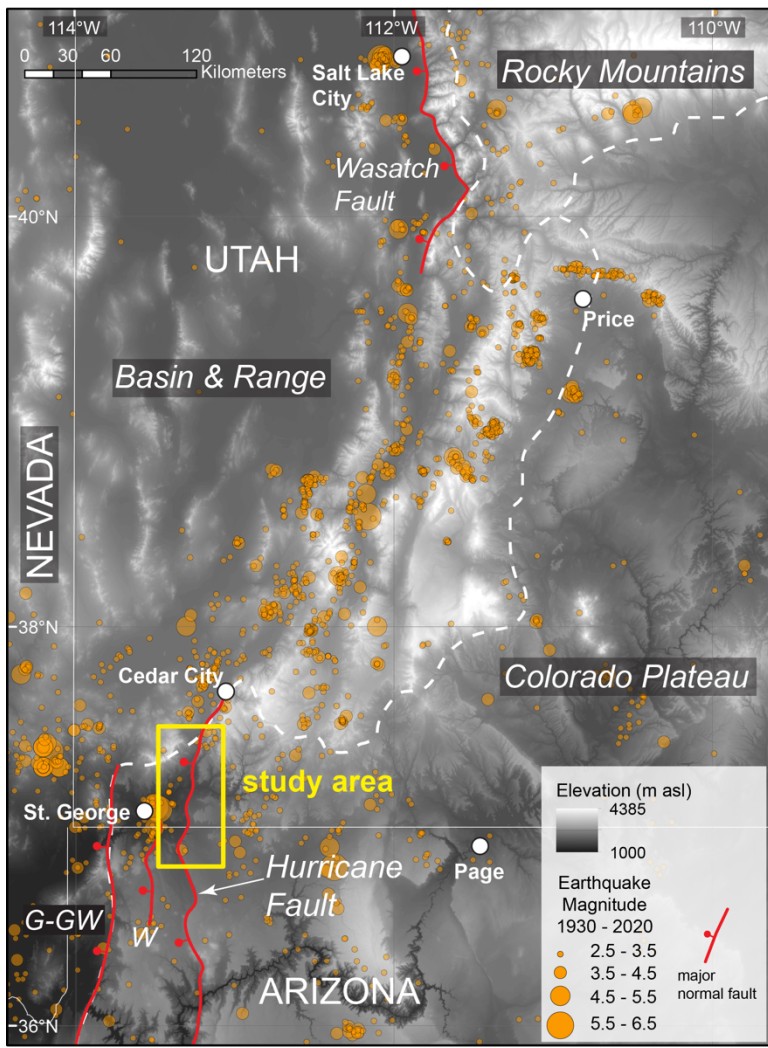

**Figure 1:** Location of the Hurricane fault and study area along the boundary between the Colorado Plateau and Basin and Range tectonic provinces in the western U.S. The fault is located within the Intermountain Seismic belt as delineated by the depicted >M 2.5 earthquakes recorded between 1930 – 2020 (USGS, 2020). Other notable faults in the region include the Gunlock-Grand Wash (G-GW) and Wasatch faults. (Digital Elevation, SRTM 1 Arc-Second Global 10.5066/F7PR7TFT, courtesy of the U.S. Geological Survey)

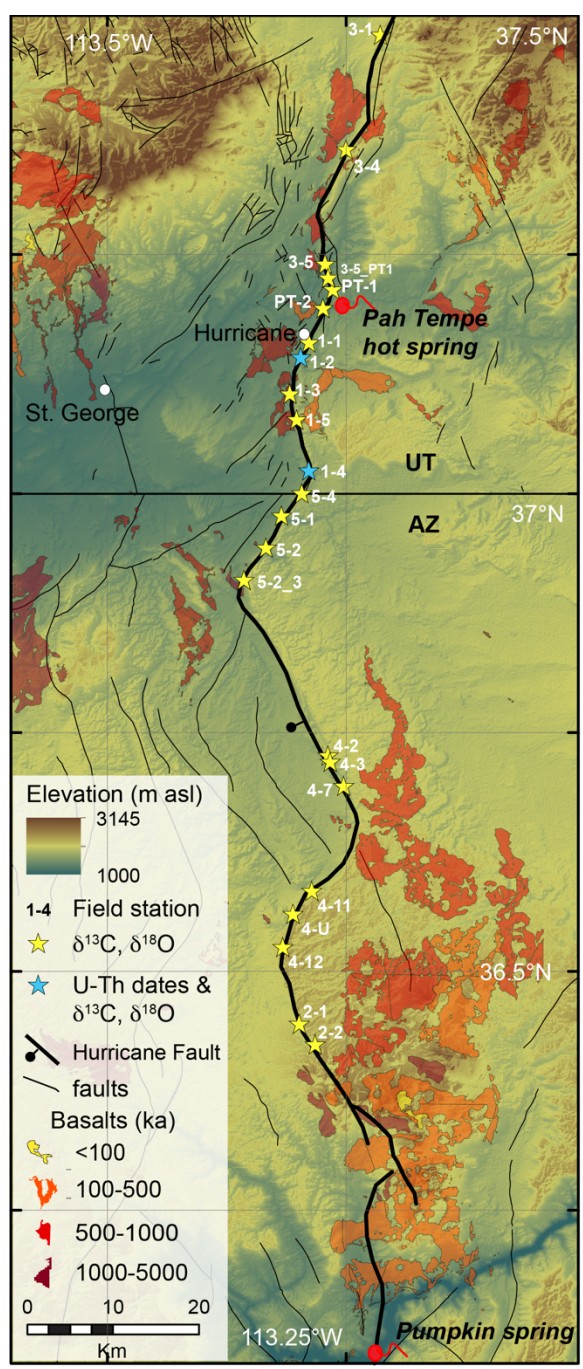

**Figure 2:** The Hurricane fault extent in southern Utah and northern Arizona with the 23 Field Stations investigated in this study. Note that C and O isotope values from calcite veins are reported for all field sites. Additionally, U-Th dates are reported from stations 1-2 and 1-4. Locations of Pah Tempe and Pumpkin springs are shown. Travertine Grotto is located south of map extent. Geology from (Billingsley and Workman, 2000; Billingsley and Wellmeyer, 2003; Rowley et al., 2008). (Digital Elevation, SRTM 1 Arc-Second Global 10.5066/F7PR7TFT, courtesy of the U.S. Geological Survey)



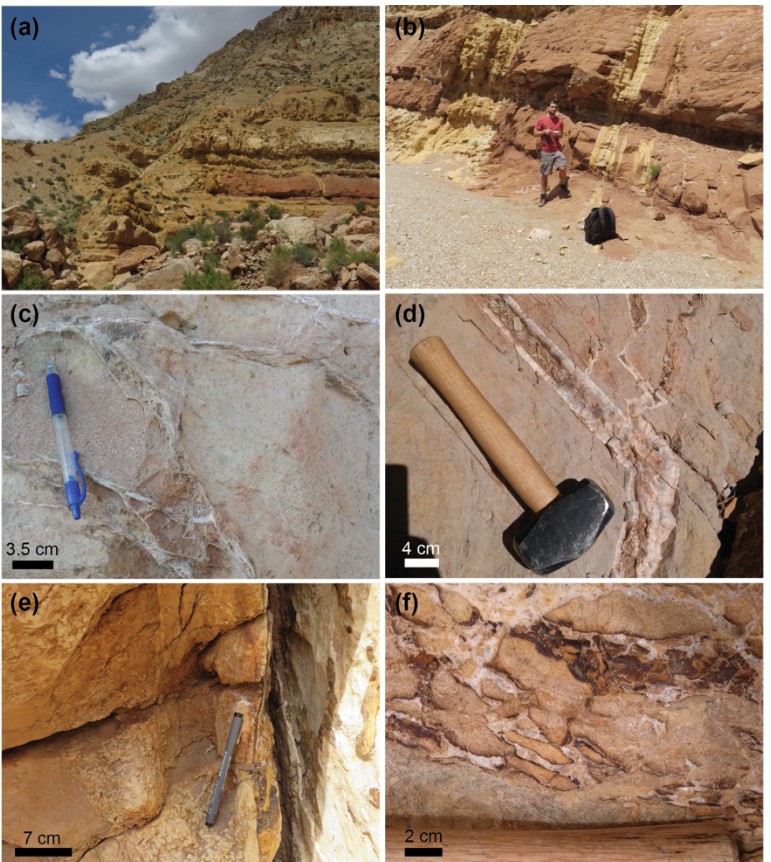

**Figure 3. (a)** Looking north along Hurricane fault trace with colluvium in the hanging wall offset against the Permian Queantoweap Sandstone. Note the bleaching along the fault trace and in horizontal strata of the Queantoweap Sandstone; **(b)** Decimeter to meter-scale bleached fractures cutting Queantoweap Sandstone, J. Koger for scale; **(c)** Boxwork sparry calcite veins **(d)** Laminated, cm-scale calcite vein with minor intergrown hematite – cutting cherty limestone host rock **(e)** Laminated calcite vein with minor hematite cutting Queantoweap Sandstone. Note the small calcite concretions cementing the sandstone parallel to the vein trace; **(f)** Calcite and hematite cemented breccia in sandstone host rock.



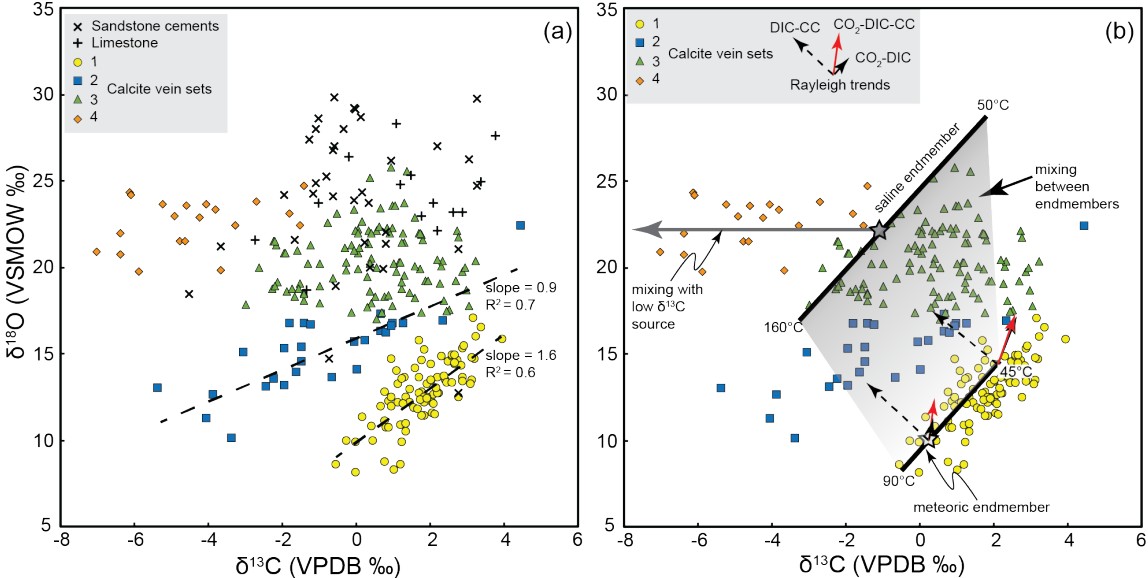

**Figure 4. (a)** Calcite $\delta^{18}O_{VSMOW}$ and $\delta^{13}C_{VPDB}$ values for veins and host rocks along the Hurricane fault. Host rock values are from bulk limestone samples and from calcite cemented sandstones. The veins are divided into 4 veins sets for analysis. Note the trend line slopes for vein sets 1 and 2 **(b)** Paleofluid interpretations integrating the isotopic and fluid inclusion microthermometry results. Mixing scenarios depicted include the mixing of two endmember fluids over a range of temperature and the mixing with a low $\delta^{13}C$ $CO_2$ source. Also shown are the Rayleigh fractionation trends (arrows) due progressive precipitation of calcite from water dissolved inorganic carbon (DIC-CC), progressive $CO_2$ loss from the water dissolved inorganic carbon (CO2-DIC), and the combined effects of $CO_2$ degassing and calcite precipitation (CO2-DIC-CC). The Rayleigh trends are computed for F from 1 to 0 (e.g., Eq. 4).



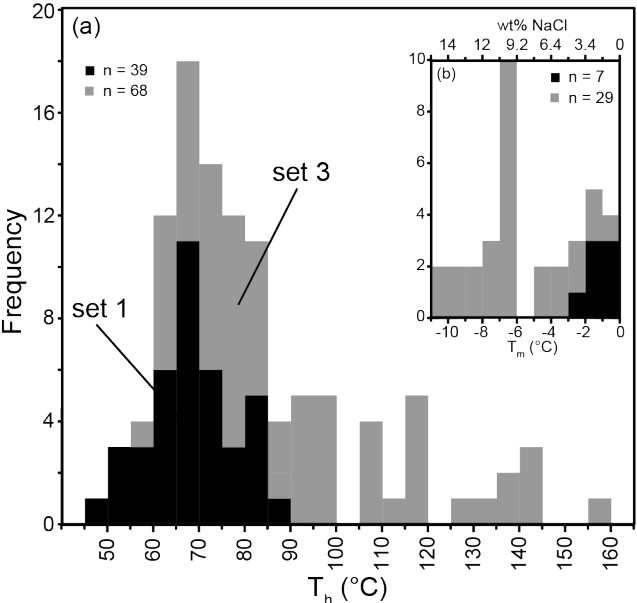

**Figure 5: (a)** Fluid inclusion (2-phase) homogenization temperatures from vein set 1 and 3. **(b)** Fluid inclusion melting temperatures from vein sets 1 and 3 and the calculated salinity as wt% NaCl (see text for details).

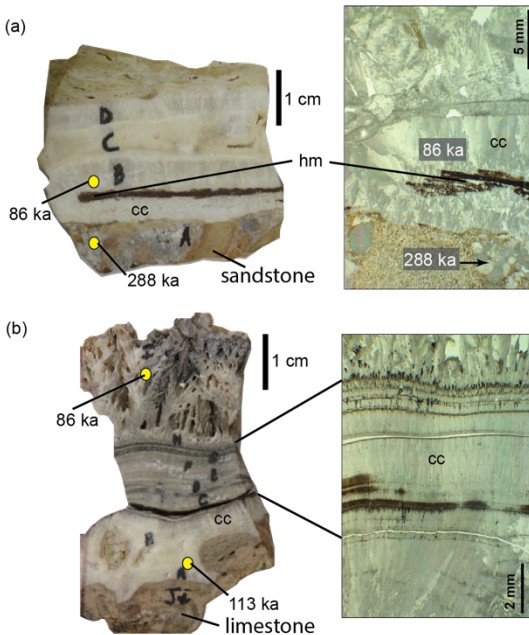

**Figure 6: (a)** Laminated calcite vein from location 1-4 and associated U-Th dates (samples JK15HR110 and JK15HR111). Hand specimen (left) and plane-polarized photomicrograph (right) shown. Note the calcite cemented brecciated sandstone forming the vein wall. The laminated calcite vein shows at least 4 episodes of calcite precipitation. The calcite cement in the wall breccia is 288 ka and the first lamination growing on the vein wall is 86 ka. **(b)** Laminated calcite vein from location 1-2 that is hosted in marine limestone. U-Th dates (JK15HR27 and JK15HR35) are shown on the hand specimen. The dates indicate growth outward from the limestone wall from 113 ka to 86 ka. Multiple dense laminations are visible in hand sample, and the plane polarized photomicrograph shows these are constructed of fibrous calcite crystals that terminate at discrete boundaries. The outermost (86 ka) layer is characterized by higher porosity vuggy calcite crystals suggestive of growth into free fluids.

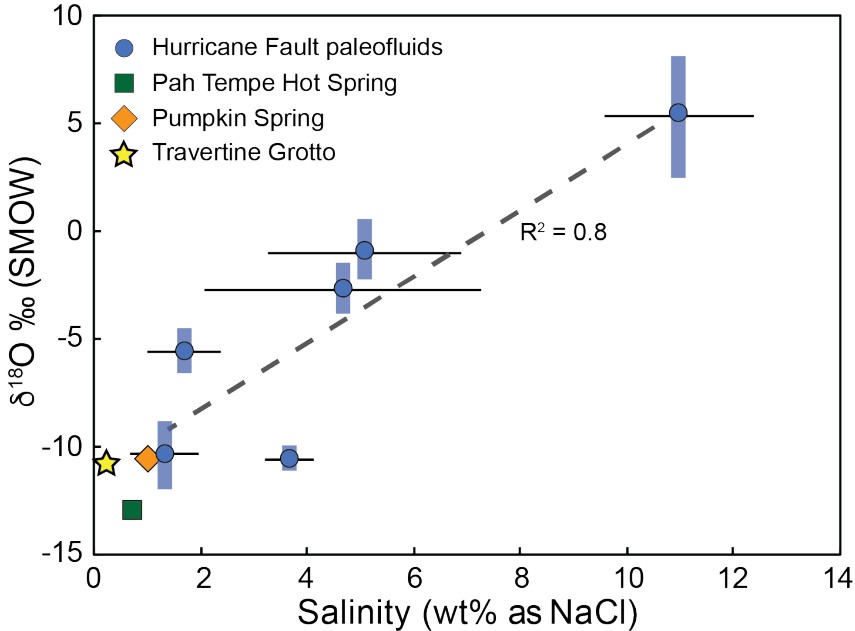

**Figure 7.** Calculated paleofluid oxygen isotope composition versus fluid salinity determined by fluid inclusion microthermometry. The strong positive correlation of $\delta^{18}O$ and salinity ($R^2 = 0.8$) is interpreted as a mixing trend between and low salinity, meteoric water and high salinity sedimentary brine endmember. For comparison, the composition of Pah Tempe, Pumpkin, and Travertine Grotto springs are included, are very similar in composition to the low salinity endmember.

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
