# Peer review of "Spatiotemporal history of fault-fluid interaction in the Hurricane fault, western USA"

_Solid Earth, 2020_

## Short Comment (SC1) · 1 Jun 2020

Dear Authors, your paper is really interesting, scientifically sound, and gives new insights into fault-fluid interaction in a fault hosting also active hydrothermal discharge. This is the best location where compare geochemical records of fossil fluids in veins with geochemical data from active springs and travertines.

I read your paper and I would like to suggest two points: -There are really good photos of outcrop structures of the Hurricane fault zone and mineralizations in the supplementary material. As in section 4.1 of the main text you describe in detail these structures, I suggest moving some of the figures from the supplementary material into the main

text. -You describe also microstructures observed on thin sections, maybe it could be worth adding some photographs of thin sections in the main text. In section 4.1 the description is detailed bu not supported by documentation in the main text.

Overall, the discussion is straightforward and the final model is strongly supported by data. Finally, I would like to draw your attention to a paper (attached) that we recently published in EPSL where we dated (U-Th) calcite veins in a normal fault in Italy and proposed a fluid-circulation model comparable to that of the Hurrican fault zone. We found very young ages (Pleistocene) of veins precipitated at shallow depth and geo-chemical analyses (stable and clumped isotopes, and noble gases analyses) indicate the uprising of deep brines, mixing with meteoric fluids, and calcite precipitation at shallow depth (< 500 m). I hope that this can further strengthen your findings.

Thanks for your contribution.

Sincerely, Luca Smeraglia

Please also note the supplement to this comment:
https://www.solid-earth-discuss.net/se-2020-69/se-2020-69-SC1-supplement.pdf

[Figure]

**Supplement:**

[supplement omitted: unrelated document]

---

## Referee Comment (RC1) · Matthew Steele-MacInnis (Referee) · 21 Jun 2020

I read the paper by Koger and Newell with interest. In my opinion, the motivation is clearly articulated, the data appear robust, and the interpretations seem sound. I recommend publication with only minor revisions.

Main comment:

My only real "main" comment is related to the origin of the saline brine. Around lines 320-326, the authors suggest that the brine originated as meteoric water, which circulated deep and acquired a high solute load. Perhaps. But there seem to be other

possibilities, and I'm not sure why they are not discussed. If the source of the salinity is thought to be marine sediments, then why shouldn't we consider paleo-seawater-derived brine as a possibility? In many cases, halogen compositions of basinal brines show evidence of salinity acquired by partial evaporation of seawater. I'm not saying this is the case here; just that it could be permissible, as far as I can tell. The authors may wish to check the papers by Bruce Yardley on this subject, and may also wish to expand the discussion of where these brines may have originated. Or, if other lines of evidence argue against something like this, then please explain that here?

And related to this previous point, a couple smaller comments:

1) The salinity of ∼11 wt% NaCl is on the low end for basinal brines. This might actually be an (equivocal) argument in favor of the brine representing original meteoric water that has picked up some solutes, though I would be wary of over-interpreting this. Basinal brines generally show salinities from ∼5 to >30 wt%, and our large dataset from MVT deposits shows a prominent mode around 20 wt% (Bodnar et al., 2014, TOG). ∼11 wt% is certainly permissible for a basinal brine, just it might be worth noting that such brines can be more saline, and this may even suggest that the true basinal "end member" has not been sampled here.

2) Basinal brines are commonly enriched in Ca, which gives rise to first-melting temperatures around -50°C. Was first melting truly never observed in this study? That is a bit unfortunate, though I guess "it is what it is." Still, I would ask you to revisit your notebooks and have a look for any notes you may have made about first melting, even if only for a few inclusions. Also, calcic brines commonly show a characteristic "orange peel" texture when frozen (Schlegel et al., 2012). Was anything like this observed?

These two latter comments are obviously little things, not crucial, but might help bolster your arguments about the brine and fluid mixing.

Detailed comments:

L11: constrains

Around L170: I suggest adding a sentence or two explaining that stretching of the fluid inclusions should have no effect on the measured Tm,ice, because stretching does not modify the composition. BUT, if the inclusions underwent any degree of leakage, then this would render the observed melting T's uninterpretable (owing to unknown degrees of H2O loss, which previous studies have shown to occur preferentially when inclusions partially leak). Hence, I assume that you did careful petrographic examination to confirm that there was no evidence of partial leakage. This should probably be stated.

L220: A very minor comment, but it is awkward phrasing to say that "Secondary minerals include primarily..." I suggest to re-word

L261: I do not understand this sentence: "Where present, single-phase fluid inclusion aperture is <15 $\mu$m." Please rephrase and clarify.

L265: "are generally inferred as <50 °C" – This is a bit misleading. Nucleation of vapor bubbles in high-density inclusions definitely depends on inclusion size (smaller inclusions are more likely to be monophase), and even inclusions with nominal Th as high as ~150°C sometimes fail to nucleate bubbles. The relationship with inclusion size should be noted here, and I would shy away from setting a rigid threshold at 50°C.

I would delete Eq2 and the sentence that precedes it. Just say that salinity was calculated using the equation of Bodnar '93.

L305: are used to estimate (not "are used to estimates of")

Around line 315: This is nice – the crux of the paper.

Around line 325: See my "main" comment above.

L327: Personally, I would use the term "meteoric water," instead of "meteoric groundwater." Simply because the term "groundwater" is sometimes used interchangeably with basinal or connate water. I'm not advocating for that (I find that even more confusing),

but just for clarity and to avoid confusion, why not "meteoric water?"

Around line 425: Don't your fluid inclusion observations provide an additional argument against a role of $CO_2$ degassing? Because of course, if $CO_2$ degassing was occurring, you ought to find vapor-rich inclusions dominated by $CO_2$. From what I can tell, there is no evidence of free $CO_2$ in your dataset, right? Maybe worth mentioning.

---

## Referee Comment (RC2) · Billy Andrews (Referee) · 15 Jul 2020

Dear Authors,

I thoroughly enjoyed reading your manuscript, which provides a detailed geochemical and geochronological study into the fault-fluid interactions of an active terrane bounding fault. In general, the data is robust, well presented, in placed within the scientific literature. In particular, Sections 5.1 & 5.2 provide nice summaries of your results, their implications, and represents the standout part of the MS. It was also refreshing to see a clear presentation of the uncertainties behind the presented conceptual models. While I have a number of minor concerns regarding the contextualisation of the presented

data, in particular how this relates to the structural relationships, this manuscript will be of great interest to the readership of Solid Earth. I therefore suggest the acceptance of the manuscript pending minor revisions. Please find my detailed comments and suggested text edits on the annotated PDF and a summary of my major and minor comments below. Kind regards, Billy J Andrews

Major comments

MC1: Lack of structural data and the field context of the samples.

A large omission from the manuscript is structural data from the different field sites and the structural relationship of the described features. This makes it difficult to place the geochemical analysis into a field setting. Wide-angle field photographs, and the inclusion of some of the field photographs in the supplementary information would greatly aid in this. I was surprised no fault or vein data was presented, either in the supplementary information or as a stereographic projection associated with the geological history of the fault. Your geochemical analysis is fantastic, but you quickly lose context without linking this to the observed structural relationships. You highlight, and I strongly agree, that the structural diagenesis, and in particular field relations and timing of these events is fundamental to the geochemical analysis. It is clear from your supplementary information, methodology section, and in part the results that this has been considered during fieldwork, However, when reading the manuscript I was often left to read between the lines, or search out images in the supplementary information, to work out what this looks like. Please consider further elaborating on the structural relationships and moving some field photographs from the supplementary information into the main text so the reader has context to the geochemical analysis.

MC2: Confusing age relationships & unclear fracture attributes (Section 4.1).

I found the vein and fracture data presented in section 4.1 rather confusing, something that was not aided by the already highlighted points in MC1. You make reference to fracture density on line 257, however, I found it very unclear how you calculated this
and whether it referred to P10 that is most often referred to as fracture intensity (f/m), or P20 that is more often referred to as fracture density. If this data was collected using scanline methods, what was the length/radius/area of the scanlines as this can have a very large impact on the reported values. If not how was the reported 'densities' calculated? This will be compounded by the fact you appear to have several fracture corridors, as suggested in line 226 where P10 will locally drastically increase.

Regarding the reported 'orientation sets' I have some the following specific questions: 1, How do the orientation sets relate to the age sets? i.e. are there systematic cross cutting relationships or are both orientations reactivated throughout the 4 stages derived from the geochemistry? 2. Where has the strike and dip data been derived from and why is strike so clustered when the fault trace at a map scale is so variable? Does the presented data represent the mean of a larger sample set and if so how many datapoints were collected? This would provide confidence that the heterogeneity of the system had been captured. Additionally, I don't understand how a dip can be 90 + 20 as the maximum dip is 90. I would like to see this data presented in the manuscript, potentially in stereonet form associated with the map? 3. I understand the more detailed field relationships were included in the supplementary information due to the focus of the paper, however, only the keenest readers will delve into this and you risk the context being lost to the majority of your readership. I suggest adding a paragraph to the main text that briefly summarises the supplementary information.

Line by line comments

Throughout the text: Please can you be consistent with the capitalisation and name of the fault. Within this paragraph is it referred to as "Hurricane fault-zone", "Hurricane Fault", and "Hurricane fault zone". Due to the segmentation I would suggest fault-zone is most appropriate. L8: I've always preferred 'fault-fluid' as the deformation is required to localise the fluid flow, with fluid impacting later deformation. L12: I've always preferred 'fault-fluid' as the deformation is required to localise the fluid flow, with fluid impacting later deformation. L16: How are these differentiated as there errors overlap? L17: All the data is taken from the FW, if possible in the word limit I would be explicit in this sentence. L31: I fully agree but It may be worth citing a few of the seminal texts, or particular pertinent texts to this study, here to direct interest readers. L32-62: This paragraph provides an important regional context that is also really important for introducing the concepts developed in the MS. I would however suggest that you split the paragraph into two- either one for each case study or a paragraph to highlight the value of studying fault-systems with ongoing fluid flow due to the tight age restraints and well defined structural evolution. L35: With the basin-bounding nature of the Hurricane-fault I think it would be worth directing readers to work such as Johnathan Caine's work on the Dixie fault (cited in this MS) and some of the work coming out of Bergen from NW Greenland (e.g. the Pre-print in this SI -> Salomon, E., Rotevatn, A., Kristensen, T. B., Grundvåg, S.-A., Henstra, G. A., Meckler, A. N., Gerdes, A., and Albert, R.: Fault-controlled fluid circulation and diagenesis along basin bounding fault systems in rifts – insights from the East Greenland rift system, Solid Earth Discuss., https://doi.org/10.5194/se-2020-72, in review, 2020. And references therein L66: This sentence is incomplete/an amalgamate of two different sentences? L66-67: please include the lithologies and what is exposed in the HW and FW of the fault. L67: please be specific here, particularly for reader who are not familiar with the geology of W. USA L75-78: This reads more like and abstract and could be removed from the introduction. Instead I suggest that you signpost the specific research gap you hope to fill. Maybe something like "Our data enables us to constrain the source and ∼540 ky evolution of fluid flow and fault-fluid interactions within the footwall of the Hurricane Fault-zone." L146: How was the degree of representation assessed? I struggled a little with understanding the outcrops from the main text alone and feel the main text sorely misses the context of field photographs, of which there are some very nice ones in the supplementary information. I strongly suggest that some of these are moved into the main text, and potentially one or two wider angle photographs included to help with contextualising the presented data. With only 6 figures and nice short nature of the MS I see no issue with adding another Figure. L215: Also structural diagenesis after Laubach et al., 2010 L218: best preserved or do you think it was localised within these competent lithologies? i.e. did diagenetic mechanical stratigraphy play a role in the fluid flow evolution of the fault zone? L217: How is damage zone and fault core constrained? I know this is not the main crucks of the paper, however, the distance that samples were taken from the fault core could impact the extracted data and overall interpretations (probably only minor in this case looking at the presented data). For reference on considering the thickness of fault zone and the bias can i suggest Shipton et al., 2019 doi: https://doi.org/10.1144/SP496-2018-161 L245: What about structural relationship? L246: Is there any visually difference between the sets? The geochemistry story is really well presented in this MS, however, I am struggling to link this in with the observed structural relationships. L256-257: What about uncertainty due to fluid degassing? L261: What does aperture refer to, short axis? You mention long axis in the previous sentence, how elongate are the fluid inclusions? L265: This depends on inclusion size, but in general i agree L279: Do you have a field photograph of this you could add to the supplementary or main text? L317: this is fairly low for saline fluids, could they be saline influenced meteoric fluids? L323: How appropriate is this for a terrane bounding setting that has had extensive volcanism? Do you have any constrains from well data? I would expect an elevated geothermal gradient. L392: One thing that may be worth signposting in the introduction is the high resolution of dates that can be obtained through the study of these systems (and hence why studies such as this are so important to the community) L441: I struggled to assess how robust this was from the presented data. The four geo-chemical set's is clear but how this fits into the field relationships is ambiguous. You only mention 2 'orientation-sets', how does cross cutting relate to these? L449: This is strongly suggested through your vein micro-structure you have presented both in the main text and the supplementary information.. this is a larger dataset and backs up the smaller dataset. I think it is worth highlighting this. L454: Is it feasible to have no exhumation of the footwall? I am not sure i agree, particularly with the differential elevation observed in Fig1. L456: see point about geothermal gradient in the previous section.. does this also have implications for the published estimates at Pah Tempe? L463: Mineralised breccias can also form due to rapid burial & differential fluid column height, see Peacock et al., 2019 -> https://doi.org/10.1111/ter.12371 L483: The shallow nature of this is a key point to highlight, although it will be a low estimate due to exhumation & erosion of the FW L491: This will have strongly effected the flow properties of the system. L523: What is the grey-scale range? could you add a scale for this in the top left of the image? F2: I would like to see the lithologies other than the basalts either in a stratigraphic column or in the presented map. Could a colour scale for elevation be added to the figure? F3: Generally really nice figure, however, a couple of suggested edits: 'calcite veins sets' appears to be slightly rotated? How are these slopes calculated? There appears to be a lot of scatter. What is the uncertainty in slopes? F4: (inset) It appears the frequency does not match the presented n values? Is there not 7 results presented for set 1 & 37 for Set 3. F5: The text size for the lithologies are too small

Supplementary information

FS1: (1) What are these two EW trending black lines referring to? The state boundary? (2) The text size for the segments are inconsistent (3) is the fault trace truly contentious? FS2: (1) Section boundary out of alignment with the figure below, i suggest shrinking the formation column slightly to give more space for the text in the member column, (2) At several points the variable text size impacts the readability of the figure. Additionally "THICKNESS" and "LITHOLOGY" should not be in full capitals. Being slightly unfamiliar with the local geology I'd have liked to the stratigraphic column in the main text to aid broader context. It could be combined with Figure 2? (3) The schematic log needs a key & grain size scale. (4) Capitalisation is missing for several Geological members (e.g. Upper Red Member) L13-14: This sentence is a little clunky, consider revising L14: Could you please present the kinematic data for the described structures either in the main text or supplementary information? How many sets and what type of sets (age, chemical, orientation)? L15: Please also check the fault name is consistent in the supplementary information and figures. L16: what proportion of the samples/studied structures? L31: How continuous are the fault breccias? L36: what is the spatial distribution of these cemented breccias relative to the main fault? L39: It would be good to see the lineation data preserved on these. Is there any variability between layers? purely extensional or is there a dip-slip component? How variable is the kinematic data across the different sites? FS3: This figure is nice and provides some of the structural context that was missing in the main text. However, could the orientation of the field photographs please be included in the figure. For clarity a scale bar could be useful (or a mention of the length of your scales in the figure caption). Also the lettering needs to be aligned with each other. In (f) mineralisation appears to be tracing along pre-existing structures here. I think a clear differentiation between 'age' sets defined by geochronology and geochemistry and 'orientation sets' needs to be woven into the manuscript. FS4: Please align lettering and similar to the previous supplementary figure please add in a scale bar and orientation to the field photographs L48: What is the type of fault breccia? L54: Do you have an appreciation of the relative timing of this alteration? is it recent GW circulation or related to the mineralisation? Is it preferentially related to specific fracture sets and/or orientations?

Please also note the supplement to this comment:
https://se.copernicus.org/preprints/se-2020-69/se-2020-69-RC2-supplement.pdf

—————————————————————

[Figure]

**Supplement:**

[revised manuscript text omitted]

---

## Author Comment (AC3) · 8 Sep 2020

Thank you for providing a comment and some suggestions on our paper. As you have suggested, we have included the suggested paper (Smeraglia et al., 2018) in our discussion. We found this paper very interesting and insightful, as well as quite consistent with what we interpret for the Hurricane fault.

Also, based on your comment as well as from another reviewer, we have moved some of the images from the supplemental information into the main text (as a new figure) to provide context.

---

## Author Response (AR1)

GEOSCIENCES

September 8, 2020

*Solid Earth*

Dear Dr. Eichhubl,

We are pleased to submit for your consideration our revised manuscript "Spatiotemporal history of fault-fluid interaction in the Hurricane fault, western U.S." by Jace M. Koger and Dennis L. Newell. The two reviews were very constructive, and both recommended minor revisions. We have incorporated all of the major suggested changes. Below we provide a detailed response to each comment and how we have changed the manuscript text and figures. We feel that the revised manuscript is much stronger, and we truly appreciate the time the reviewers spent evaluating our work.

In addition to the revised manuscript and supplemental, we have included with this document, pdf's of the track-changes versions of the main text and supplemental.

Thank you for considering our work for publication in *Solid Earth*!

Best regards,

Dennis Newell

Corresponding Author

4505 Old Main Hill
Logan, UT 84322-4505
P-435-797-1273
F-435-797-1588
geo@usu.edu

[Figure]

In the following we detail how we have addressed each of the comments and suggestions from Billy Andrews and Matthew Steele-MacInnis. In addition, some edits were made to the manuscript pdf file by Billy Andrews, and these have been incorporated into the revised manuscript.

**Comments and suggestions from Billy Andrews**

**Major comments**

**MC1: Lack of structural data and the field context of the samples.**

A large omission from the manuscript is structural data from the different field sites and the structural relationship of the described features. This makes it difficult to place the geochemical analysis into a field setting. Wide-angle field photographs, and the inclusion of some of the field photographs in the supplementary information would greatly aid in this. I was surprised no fault or vein data was presented, either in the supplementary information or as a stereographic projection associated with the geological history of the fault. Your geochemical analysis is fantastic, but you quickly lose context without linking this to the observed structural relationships. You highlight, and I strongly agree, that the structural diagenesis, and in particular field relations and timing of these events is fundamental to the geochemical analysis. It is clear from your supplementary information, methodology section, and in part the results that this has been considered during fieldwork, However, when reading the manuscript I was often left to read between the lines, or search out images in the supplementary information, to work out what this looks like. Please consider further elaborating on the structural relationships and moving some field photographs from the supplementary information into the main text so the reader has context to the geochemical analysis.

Author Response:

We agree that the omission of details associated with the structural data detracts from the interpretation of the geochemical results. To rectify this, the following changes have been made.

1) A new supplemental figure (map) has been added that includes stereonets from all the field sites. The stereonet data is color-coded to show the different vein sets and measured fractures. In total these stereonets include 477 orientation measurements.

**MC2: Confusing age relationships & unclear fracture attributes (Section 4.1).**

I found the vein and fracture data presented in section 4.1 rather confusing, something that was not aided by the already highlighted points in MC1. You make reference to fracture density on line 257, however, I found it very unclear how you calculated this and whether it referred to $P_{10}$ that is most often referred to as fracture intensity (f/m), or $P_{20}$ that is more often referred to as fracture density. If this data was collected using scanline methods, what was the length/radius/area of the scanlines as this can have a very large impact on the reported values. If not how was the reported 'densities' calculated? This will be compounded by the fact you appear to have several fracture corridors, as suggested in line 226 where $P_{10}$ will locally drastically increase.

Author Response:

We measured the fracture intensity (f/m) and incorrectly called this fracture density. The text is changed to make this clear. Also, we used a linear scanline method, using a tape measure oriented approximately orthogonal to the main fault trace. For the footwall outcrops observed along the Hurricane fault, this worked well as most observed fractures are intersected this way. We recognize that there can be bias using this simple method, but for the field sites visited this was the most efficient method. The length of the scanlines was variable and based on the available exposures. Drainages and canyons that cut the fault and run approximately orthogonal to the fault were the focus of the data collection due to the best exposures. Some canyons are relatively long (> 1 km), allowing the transect to traverse the damage zone completely. Other small drainages only penetrate part of the damage zone. Yes, as you point out there are some fracture corridors in the damage zone that have greater fracture intensity. We have updated the text to make it clear how the data was collected. Specifically, details of the methods were added to section 3.1, and section 4.1 and the supplement were updated with the correct terminology.

Regarding the reported 'orientation sets' I have some the following specific questions:

1, How do the orientation sets relate to the age sets? i.e. are there systematic cross cutting relationships or are both orientations reactivated throughout the 4 stages derived from the geochemistry?

Author Response:

In a very general sense, there seems to be systematic cross-cutting relationships along the fault. However, we downplay this in the paper given the limited geochronological data set. Locally, where we have U-Th data, the cross-cutting relationships hold up (e.g., Fig 5). Using the geochemistry and geochronology, to a first order it appears that veins associated with the lower salinity fluids with a strong meteoric water affinity are older (including the 280 – 540 ka veins), compared to the saltier fluids characterized in the 86 – 113 ka veins (Fig. S7). In order to really flesh this out and present a more comprehensive paragenetic history linked to vein set, significantly more geochronological work is necessary. Jace Koger's MSc thesis, which is the foundation of this paper, takes a general crack at the paragenetic history based on field and microscopic relationships, but we prefer to hold this back until more U-Th work is completed to anchor the observations.

2. Where has the strike and dip data been derived from and why is strike so clustered when the fault trace at a map scale is so variable? Does the presented data represent the mean of a larger sample set and if so how many datapoints were collected? This would provide confidence that the heterogeneity of the system had been captured. Additionally, I don't understand how a dip can be 90 $\pm$ 20 as the maximum dip is 90. I would like to see this data presented in the manuscript, potentially in stereonet form associated with the map?

Author Response:

The fault trace shown on the figures is from the Utah and Arizona Geological Survey's, available as ArcGIS layers on their websites. These databases are based on geological mapping at a variety of scales. Exposures of the main trace of the Hurricane Fault are not common and usually covered with colluvium. The fracture's that we measured are found close to the trace of the fault and are in general consistent with the map pattern, with some exceptions and variability. To depict this, a new figure (S3) is provided in the supplement that presents stereonets for each field station. This should help show how the facture patterns and fault orientation are generally similar.

Yes, 90 +/- 20 does not make any sense! This has been changed to 70 – 90 degrees.

3) I understand the more detailed field relationships were included in the supplementary information due to the focus of the paper, however, only the keenest readers will delve into this and you risk the context being lost to the majority of your readership. I suggest adding a paragraph to the main text that briefly summarises the supplementary information.

Author Response:

We have added additional details to the main text (section 4.1) that includes important contextual details from the supplemental material. Additionally, a new figure is included in this section.

**Line by line comments**

**Throughout the text:** Please can you be consistent with the capitalisation and name of the fault. Within this paragraph is it referred to as "Hurricane fault-zone", "Hurricane Fault", and "Hurricane fault zone". Due to the segmentation I would suggest fault-zone is most appropriate.

Author Response: We agree that different notation is used and that this is distracting. Looking at the various Utah and Arizona Geological Survey reports on the fault, as well as the published literature (e.g., Stewart and Taylor, 1996), it appears that the most consistent usage is "Hurricane fault". We agree that fault zone is more accurate in most locations. We have altered the text as appropriate.

**L8:** I've always preferred 'fault-fluid' as the deformation is required to localise the fluid flow, with fluid impacting later deformation.

Author Response: Changed to fault-fluid throughout manuscript, including the title.

**L16:** How are these differentiated as there errors overlap?

Author Response: Here in the abstract we are just reported the five dates and their 2 sigma errors. Later in the paper we make the argument that these are the same fluid flow event.

**L17:** All the data is taken from the FW, if possible in the word limit I would be explicit in this sentence.

Author Response: We added this information

**L31:** I fully agree but It may be worth citing a few of the seminal texts, or particular pertinent texts to this study, here to direct interest readers.

Author Response: We indicate that these will be highlighted in the next section

**L32-62:** This paragraph provides an important regional context that is also really important for introducing the concepts developed in the MS. I would however suggest that you split the paragraph into two- either one for each case study or a paragraph to highlight the value of studying fault-systems with ongoing fluid flow due to the tight age restraints and well defined structural evolution.

Author Response: Excellent suggestion, and we have divided the section into paragraphs

**L35:** With the basin-bounding nature of the Hurricane-fault I think it would be worth directing readers to work such as Johnathan Caine's work on the Dixie fault (cited in this MS) and some of the work coming out of Bergen from NW Greenland (e.g. the Pre-print in this SI -> Salomon, E., Rotevatn, A., Kristensen, T. B., Grundvåg, S.-A., Henstra, G. A., Meckler, A. N., Gerdes, A., and Albert, R.: Fault-controlled fluid circulation and diagenesis along basin bounding fault systems in rifts – insights from the East Greenland rift system, Solid Earth Discuss., https://doi.org/10.5194/se-2020-72, in review, 2020. And references therein

Author Response: These and other references have been added

**L66:** This sentence is incomplete/an amalgamate of two different sentences?

Author Response: Yes, this was a broken sentence and has been corrected.

**L66-67:** please include the lithologies and what is exposed in the HW and FW of the fault.

Author Response: This was added.

**L67:** please be specific here, particularly for reader who are not familiar with the geology of W. USA

Author Response: See prior response

**L75-78:** This reads more like and abstract and could be removed from the introduction. Instead I suggest that you signpost the specific research gap you hope to fill. Maybe something like "Our data enables us to constrain the source and ~540 ky evolution of fluid flow and fault-fluid interactions within the footwall of the Hurricane Fault-zone."

Author Response: Thank you for the suggested sentence. We have changed the last part of this paragraph.

**L146:** How was the degree of representation assessed? I struggled a little with understanding the outcrops from the main text alone and feel the main text sorely misses the context of field photographs, of which there are some very nice ones in the supplementary information. I strongly suggest that some of these are moved into the main text, and potentially one or two wider angle photographs included to help with contextualising the presented data. With only 6 figures and nice short nature of the MS I see no issue with adding another Figure.

Author Response: The degree of representation is subjective, based on the evidence of fault-fluid interaction at each field station. This study is the first of its kind along the Hurricane fault, so the goal was to cover as much ground as possible. We could not for example use any statistical

approaches for collecting the samples. We have added a figure (new figure 3) to the main text in section 4.1 as suggested to show some of the representative field relationships and outcrop styles.

**L215:** Also structural diagenesis after Laubach et al., 2010

Author Response: Yes, this is a good suggestion. Citation added

**L218:** best preserved or do you think it was localised within these competent lithologies? i.e. did diagenetic mechanical stratigraphy play a role in the fluid flow evolution of the fault zone?

Author Response: That is a great question. In some cases it is preservation because the fracturing is evident in the siltstones and shales, but these tend to be poorly exposed. In other cases the fracturing is apparent in all lithologies, but the calcite veining is only present in the sandstone or limestone units, so in this case in could be related to mechanical stratigraphy and/or permeability contrasts.

**L217:** How is damage zone and fault core constrained? I know this is not the main crucks of the paper, however, the distance that samples were taken from the fault core could impact the extracted data and overall interpretations (probably only minor in this case looking at the presented data). For reference on considering the thickness of fault zone and the bias can i suggest Shipton et al., 2019 doi: https://doi.org/10.1144/SP496-2018-161

Author Response: These are defined using the criteria in Caine et al., 2010. At most field sites, samples were collected fairly close to the main trace of the fault. Only in a few canyons that cut deeper into the footwall could the full damage zone thickness be walked and sampled. In these cases, clear evidence for fracturing and fluid flow were still closest to the fault trace (~100 m), but similarly oriented fractures and veins were observed up to 400 m from the fault trace. Some clarify text was added to the paper.

**L245:** What about structural relationship?

**L246:** Is there any visually difference between the sets? The geochemistry story is really well presented in this MS, however, I am struggling to link this in with the observed structural relationships.

Author Response: In response to both the above comments. There does not appear to be any consistent relationship between the structural data and vein type. This is consistent with our later interpretation that fluids are periodically moving along the fault zone. Also, over the history of the Hurricane fault, the regional stress field has been the same, so we probably should not expect distinctly different orientations associated with the history of vein formation.

**L256-257:** What about uncertainty due to fluid degassing?

Author Response: Later we discount degassing as a major process. However, degassing of the fluid ($CO_2$) should not significantly alter the salinity from NaCl or other dissolved salts, so its impact on the melting T would be secondary.

**L261:** What does aperture refer to, short axis? You mention long axis in the previous sentence, how elongate are the fluid inclusions?

Author Response: This has been corrected. We meant long axis.

**L265:** This depends on inclusion size, but in general i agree

Author Response: We have updated this section to discuss the impact of inclusion size on nucleation of a bubble and homogenization temperature.

**L279:** Do you have a field photograph of this you could add to the supplementary or main text?

Author Response: Unfortunately, this information is only in the field notes.

**L317:** this is fairly low for saline fluids, could they be saline influenced meteoric fluids?

Author Response: We have added to this discussion about the saline fluids and their possible origin. In

short, this is on the low side for basin brines. Certainly, meteoric water mixing with a brine can result in the observed salinities, as well as meteoric water-rock interaction along flow paths through marine units that include some evaportites.

**L323:** How appropriate is this for a terrane bounding setting that has had extensive volcanism? Do you have any constrains from well data? I would expect an elevated geothermal gradient.

Author Response: There are some data from geothermal exploration wells in the region. The gradients observed are mostly in the 18 – 24 °C/km, but a couple wells yielded 34 and 175 °C/km. So it is likely variable and perhaps elevated near the fault. We suggest however that the nominal gradients are probably more appropriate for the regional groundwater flow patterns.

Sommer, S. N., and Budding, K. E.: Low-temperature thermal waters in the Santa Clara and Virgin River valleys, Washington County, Utah, in: Cenozoic Geology and Geothermal Systems of Southwestern Utah, edited by: Blackett, R. E., and Moore, J., Utah Geological Association, Salt Lake City, 81-95, 1994.

**L392:** One thing that may be worth signposting in the introduction is the high resolution of dates that can be obtained through the study of these systems (and hence why studies such as this are so important to the community)

Author Response: Thank you for the suggestion – we have added a statement to this effect at the end of the introduction.

**L441:** I struggled to assess how robust this was from the presented data. The four geo-chemical set's is clear but how this fits into the field relationships is ambiguous. You only mention 2 'orientation-sets', how does cross cutting relate to these?

Author Response: We have updated the text to be more clear about these relationships.

**L449:** This is strongly suggested through your vein micro-structure you have presented both in the main text and the supplementary information.. this is a larger dataset and backs up the smaller dataset. I think it is worth highlighting this.
Author Response: We agree.

**L454:** Is it feasible to have no exhumation of the footwall? I am not sure i agree, particularly with the differential elevation observed in Fig1.

Author Response: We agree that there must be some exhumation of the footwall. The best constraint we can apply is using the incision rate of the Virgin River as a maximum – assuming the landscape is in steady state, the erosion rate might be a good measure of this (a maximum). The Virgin River has an estimated incision rate of 338 m/Myr (Walk et al., 2019), and this equates to ~180 m of incision since 540 ka. We added this information to the text to estimate the maximum of depth vein formation – it is still shallow (70 to 480 m).

**L456:** see point about geothermal gradient in the previous section.. does this also have implications for the published estimates at Pah Tempe?

Author Response: See prior comment on the constraints on the geothermal gradient.

**L463:** Mineralised breccias can also form due to rapid burial & differential fluid column height, see Peacock et al., 2019 -> https://doi.org/10.1111/ter.12371

Author Response: Thank you for this citation. Based on the geological history of the area and this fault, we do not think this process is relevant in this case.

**L483:** The shallow nature of this is a key point to highlight, although it will be a low estimate due to

exhumation & erosion of the FW

Author Response: We agree, and we have improved the depth estimate.

**L491:** This will have strongly effected the flow properties of the system.

Author Response: Text was added to highlight this.

**L523:** What is the grey-scale range? could you add a scale for this in the top left of the image?

Author Response: An elevation scale bar was added.

**F2:** I would like to see the lithologies other than the basalts either in a stratigraphic column or in the presented map. Could a colour scale for elevation be added to the figure?

Author Response: We feel that added the lithologies to this map will be too busy and render it very difficult to read, and we refer the readers to the excellent geological maps of the area that are all available on the Utah Geological Society's website, and free to download. A color scale bar for elevation was added.

**F3:** Generally really nice figure, however, a couple of suggested edits: 'calcite veins sets' appears to be slightly rotated? How are these slopes calculated? There appears to be a lot of scatter. What is the uncertainty in slopes?

Author Response: Thank you! The slopes were determined by linear regression. Agreed, there is a fair amount of scatter, and we have added the r^2 values to the plot to help quantify this.

**F4:** (inset) It appears the frequency does not match the presented n values? Is there not 7 results presented for set 1 & 37 for Set 3.

Author Response: Correct! We fixed the figure.

**F5:** The text size for the lithologies are too small

Author Response: The text size was increased

**Supplementary information**

**FS1:** (1) What are these two EW trending black lines referring to? The state boundary? (2) The text size for the segments are inconsistent (3) is the fault trace truly contentious?

Author Response: Those E-W lines were errors in the figure and have been corrected. The text size has been fixed. To the north the Hurricane fault trace is less distinct and smaller faults step over to the East.

**FS2:** (1) Section boundary out of alignment with the figure below, i suggest shrinking the formation column slightly to give more space for the text in the member column, (2) At several points the variable text size impacts the readability of the figure. Additionally "THICKNESS" and "LITHOLOGY" should not be in full capitals. Being slightly unfamiliar with the local geology I'd have liked to the stratigraphic column in the main text to aid broader context. It could be combined with Figure 2? (3) The schematic log needs a key & grain size scale. (4) Capitalisation is missing for several Geological members (e.g. Upper Red Member)

Author Response: We have revised this figure significantly. In fact, we have simplified the figure because it included unnecessary detail that is not discussed in the paper. This figure is just to

provide some context for the readers, but it is beyond the scope of this paper to recast the excellent geological mapping that has occurred in the area and is readily available.

**L13-14:** This sentence is a little clunky, consider revising

Author Response: Fixed.

**L14:** Could you please present the kinematic data for the described structures either in the main text or supplementary information? How many sets and what type of sets (age, chemical, orientation)?

Author Response: A new figure is included in the supplement with the vein and fracture orientations.

**L15:** Please also check the fault name is consistent in the supplementary information and figures.

Author Response: The supplement was updated for consistency.

**L16:** what proportion of the samples/studied structures?

Author Response: Probably 70%

**L31**: How continuous are the fault breccias?

Author Response: It depends on the location. There are some exposures where the fault breccias are continuous over 100's of meters, and in other locations very localized.

**L36:** what is the spatial distribution of these cemented breccias relative to the main fault?

Author Response: Based on the data we have, we cannot quantify this across the whole study area. Where we have observations, the breccias are either directly associated with the main fault trace or are within several meters.

**L39:** It would be good to see the lineation data preserved on these. Is there any variability between layers? purely extensional or is there a dip-slip component? How variable is the kinematic data across the different sites?

Author Response: We only have limited data on the slip surfaces.

**FS3:** This figure is nice and provides some of the structural context that was missing in the main text. However, could the orientation of the field photographs please be included in the figure. For clarity a scale bar could be useful (or a mention of the length of your scales in the figure caption). Also the lettering needs to be aligned with each other. In (f) mineralisation appears to be tracing along pre-existing structures here. I think a clear differentiation between 'age' sets defined by geochronology and geochemistry and 'orientation sets' needs to be woven into the manuscript.

Author Response: The figure has been edited and some of this content is now in the main text. With respect the last sentence, we feel that we have differentiated between relationships constrained by geochronology versus those based on observed cross-cutting relationships.

**FS4:** Please align lettering and similar to the previous supplementary figure please add in a scale bar and orientation to the field photographs

Author Response: The figure has been modified to adjust the lettering. These are unoriented photos, and include some common objects for scale in the photos.

**L48:** What is the type of fault breccia?

Author Response: We did not classify the fault breccias we observed.

**L54:** Do you have an appreciation of the relative timing of this alteration? is it recent GW circulation or related to the mineralisation? Is it preferentially related to specific fracture sets and/or orientations?

Author Response: This is a good question. Given that the reducing fluids are needed to bleach these zones, we do not think they are related to recent groundwater circulation. This alteration is most common closest to the main fault trace, associated mostly with the ~N-S oriented fractures.

**Review by Matthew Steele-MacInnis**

I read the paper by Koger and Newell with interest. In my opinion, the motivation is clearly articulated, the data appear robust, and the interpretations seem sound. I recommend publication with only minor revisions.

Main comment:

My only real "main" comment is related to the origin of the saline brine. Around lines 320-326, the authors suggest that the brine originated as meteoric water, which circulated deep and acquired a high solute load. Perhaps. But there seem to be other possibilities, and I'm not sure why they are not discussed. If the source of the salinity is thought to be marine sediments, then why shouldn't we consider paleo-seawater- derived brine as a possibility? In many cases, halogen compositions of basinal brines show evidence of salinity acquired by partial evaporation of seawater. I'm not saying this is the case here; just that it could be permissible, as far as I can tell. The authors may wish to check the papers by Bruce Yardley on this subject, and may also wish to expand the discussion of where these brines may have originated. Or, if other lines of evidence argue against something like this, then please explain that here?

Author response:

We agree that other sources of salinity are a possibility and we only provide our preferred interpretation. Certainly, the $\delta18O$ and moderate salinity of our most saline endmember could represent some fraction of a paleo-seawater derived brine. Unfortunately, we do not have halogen data (e.g., Cl/Br) available from our samples, and these data are not published for the thermal springs along the Hurricane fault (Pah Tempe, Travertine Grotto) that could help with fingerprinting the source. Thus with the available data we cannot distinguish between meteoric water mixing with evolved paleo-seawater or meteoric-water-rock interaction.

As the reviewer points out in comment #1, the relatively low salinity (11 wt %), although within the range observed for basinal brines, is on the low end, and thus our dataset may not capture the saline endmember. We agree that our mixing trend in Figure 6 could extend to higher salinity, higher $\delta18O$ values.

We have updated our discussion to address the other possible interpretations.

And related to this previous point, a couple smaller comments:

1) The salinity of 11 wt% NaCl is on the low end for basinal brines. This might actually be an (equivocal) argument in favor of the brine representing original meteoric water that has picked up some solutes, though I would be wary of over-interpreting this. Basinal brines generally show salinities from 5 to >30 wt%, and our large dataset from MVT deposits shows a prominent mode around 20 wt% (Bodnar et al., 2014, TOG). 11 wt% is certainly permissible for a basinal brine, just it might be worth noting that such

brines can be more saline, and this may even suggest that the true basinal "end member" has not been sampled here.

Author response: See our response to the above "main comment".

2) Basinal brines are commonly enriched in Ca, which gives rise to first-melting tem- peratures around -50°C. Was first melting truly never observed in this study? That is a bit unfortunate, though I guess "it is what it is." Still, I would ask you to revisit your notebooks and have a look for any notes you may have made about first melting, even if only for a few inclusions. Also, calcic brines commonly show a characteristic "orange peel" texture when frozen (Schlegel et al., 2012). Was anything like this observed?

Author response:

During the fluid inclusion measurements, we looked very carefully for first melting, and it was not observed. Also, we were not aware of the "orange peel" texture at the time of analysis; however, we did not note any unusual textures during freezing. Perhaps future work on these samples could utilize other methods to address the actual composition of the fluid inclusions (such as Raman work).

These two latter comments are obviously little things, not crucial, but might help bolster your arguments about the brine and fluid mixing.

Detailed comments:

L11: constrains

Author response: corrected

Around L170: I suggest adding a sentence or two explaining that stretching of the fluid inclusions should have no effect on the measured Tm,ice, because stretching does not modify the composition. BUT, if the inclusions underwent any degree of leakage, then this would render the observed melting T's uninterpretable (owing to unknown degrees of $H_2O$ loss, which previous studies have shown to occur preferentially when inclu- sions partially leak). Hence, I assume that you did careful petrographic examination to confirm that there was no evidence of partial leakage. This should probably be stated.

Author response: This is a good point, and the text has been revised to include this discussion.

L220: A very minor comment, but it is awkward phrasing to say that "Secondary min- erals include primarily..." I suggest to re-word

Author response: We reworded this sentence.

L261: I do not understand this sentence: "Where present, single-phase fluid inclusion aperture is <15 $\mu$m." Please rephrase and clarify.

Author response: This sentence was rewritten for clarity.

L265: "are generally inferred as <50 °C" – This is a bit misleading. Nucleation of vapor bubbles in highdensity inclusions definitely depends on inclusion size (smaller inclusions are more likely to be monophase), and even inclusions with nominal Th as high as 150°C sometimes fail to nucleate bubbles. The relationship with inclusion size should be noted here, and I would shy away from setting a rigid threshold at 50°C.

Author response: Thank you for pointing this out; the text has been modified to include this information.

I would delete Eq2 and the sentence that precedes it. Just say that salinity was calcu- lated using the equation of Bodnar '93.

Author response: We have changed the sentence as suggested and removed the equation.

L305: are used to estimate (not "are used to estimates of")

Author response: corrected

Around line 315: This is nice – the crux of the paper.

Author response: We agree, thank you.

Around line 325: See my "main" comment above.

Author response: See above discussion.

L327: Personally, I would use the term "meteoric water," instead of "meteoric groundwa- ter." Simply because the term "groundwater" is sometimes used interchangeably with basinal or connate water. I'm not advocating for that (I find that even more confusing but just for clarity and to avoid confusion, why not "meteoric water?"

Author response: Where appropriate, we have changed meteoric groundwater to meteoric water.

Around line 425: Don't your fluid inclusion observations provide an additional argument against a role of $CO_2$ degassing? Because of course, if $CO_2$ degassing was occurring, you ought to find vapor-rich inclusions dominated by $CO_2$. From what I can tell, there is no evidence of free $CO_2$ in your dataset, right? Maybe worth mentioning.

Author response: This is a good point. We do not observe any vapor dominated fluid inclusions. We have added this detail to the discussion.

[revised manuscript text omitted]

**S2 Hurricane fault-zone fractures, veins, and alteration**

**S2.1 Fracture and vein orientations**

[Figure]

**Figure S3.** Fracture and vein orientation along the Hurricane fault-zone. Stereonets correspond to each field station and depict the veins sets and fractures measured in the footwall damage zone (stereonets include a total of 477 vein and fracture measurements).

**S2.2 Calcite veins and fracture fills**

Various morphologies of veins and fracture coatings are present along the main fault trace and in the footwall damage zone (Fig. S4). Cross-cutting veins at multiple sites indicate that multiple episodes of fracturing and mineralization occurred (Fig. S4 a). Calcite is ubiquitous as a fracture and vein mineral along the Hurricane fault zone. The majority (~70%) of veins observed are composed of sparry calcite, and are particularly common cutting limestone strata (Fig. S4 b). Interconnected, web-like "boxwork" calcite veins are common in sandstone within ~ 50 m of the main fault trace (Fig. S4 c). Other

[revised manuscript text omitted]

width with some on the order of 0.1 to 3 cm (Fig. S6 c) and others on the decimeter to meter-scale (Fig. S6 a, b). Alternating

stratigraphic horizons are also bleached separated by unaltered strata (Fig. S6 a, c). The degree of sandstone cementation by

90 calcite also changes in the fault zone. Near the fault trace, calcite cementation, including the presence of calcite concretions

is greatest adjacent to fractures, and fractures hosting calcite veins (Fig. S6 d).

[Figure]

**Figure S6.** (a) Looking north along Hurricane fault trace. Note the offset Permian Queantoweap Sandstone displaying bleaching along the fault trace and in horizontal strata; (b) Decimeter to meter-scale bleached fractures cutting Queantoweap Sandstone; (c) Densely fractured silty sandstone in the Hermit Formation showing mm- to cm-scale bleaching along fractures; (d) Bedding parallel bleaching in siltstones bounded by unbleached strata.

105

| | |
|---|---|
| **Deleted:** 5 | |
| **Formatted:** Font: Not Bold | |
| **Deleted:** F | |
| **Deleted:** Hermit Formation | |

110 **S3 Data table and figure supplements**

**Table S1. Location of 23 Field Stations along the Hurricane Fault**

| Field Station | Latitude[a] | Longitude |
|---|---|---|
| 3-1 | 37.477510 | -113.21409 |
| 3-4 | 37.361312 | -113.25263 |
| 3-5 | 37.237774 | -113.27133 |
| 3-5_PT1 | 37.227990 | -113.25961 |
| PT-1 | 37.212103 | -113.26174 |
| PT-2 | 37.191258 | -113.27319 |
| 1-1 | 37.157048 | -113.28758 |
| 1-2 | 37.137324 | -113.29661 |
| 1-3 | 37.103990 | -113.30294 |
| 1-5 | 37.080825 | -113.30608 |
| 1-4 | 37.017787 | -113.28893 |
| 5-4 | 36.996654 | -113.30263 |
| 5-1 | 36.973909 | -113.31452 |
| 5-2 | 36.942954 | -113.33335 |
| 5-2_3 | 36.925742 | -113.35111 |
| 4-2 | 36.725976 | -113.26352 |
| 4-3 | 36.714396 | -113.25492 |
| 4-7 | 36.695012 | -113.24731 |
| 4-11 | 36.580715 | -113.28505 |
| 4-U | 36.569903 | -113.29930 |
| 4-12 | 36.519993 | -113.31926 |
| 2-1 | 36.443720 | -113.29808 |
| 2-2 | 36.421994 | -113.28260 |

[a] WGS 84 datum

115

**Table S2. U-Th data for the 5 calcite veins. All ratios are activity ratios.**

| Sample ID | $^{230}Th/^{238}U_{true}$[a] | $^{234}U/^{238}U_{true}$ | $^{230}Th/^{232}Th_{true}$ | $^{232}Th/^{238}U_{true}$ | $\delta^{235}U_{rel\,to\,CRM145}$ | $\delta^{235}U_{stds\,rel\,to\,CRM145}$ | age (y)[b] | $^{234}U/^{238}U_{init}$ |
|---|---|---|---|---|---|---|---|---|
| JK15HR110 | 0.9373 | 1.007 | 1510 | 6.21E-04 | 0.0 | | 287,856 ± 5757 | |
| JK15HR111 | 0.8306 ± 0.0008 | 1.469 ± 0.002 | 588±1.23 | 1.41E-03 ± 2.85E-06 | -0.3±0.2 | -0.3±0.3 | 85,993 ± 196 | 1.598 ± 0.004 |
| JK15HR103 | 1.1609 | 1.120 | 87 | 1.33E-02 | -0.1 | | 539,304 ± 10786 | |
| JK15HR27 | 0.9437 ± 0.0009 | 1.403 ± 0.002 | 240±0.503 | 3.94E-03 ± 7.95E-06 | 0.5±0. | 0.0±0.3 | 113,062 ± 315 | 1.555 ± 0.004 |
| JK15HR35 | 0.8110 | 1.435 | 422 | 1.92E-03 | 0.0 | | 86,233± 1725 | |

[a] Where reported, errors are 1-sigma. Due to slight method differences, these errors are not available for all samples
[b] Errors for JK15HR110 and 103 are estimated as 2%, rather than 1-sigma due to method differences

120

[Figure]

Figure S7. Carbon and oxygen stable isotope values and corresponding U-Th dates for the 5 dated samples. The likely fluid-endmember is identified based on stable isotope values.

**References Cited**

125   Biek, R.: Geologic Map of the Hurricane Quadrangle Washington County, Utah: Utah Geological Survey Map, 187, 2003.
Biek, R., Rowley, P., Hayden, J., Hacker, D., Willis, G., Hintze, L., Anderson, R., and Brown, K.: Geologic map of the St. George and east part of the Clover Mountains 30'X60' quadrangles, Washington and Iron counties, Utah, Utah Geological Society, 2010.
Dutson, S.: Effects of Hurricane Fault Architecture on Groundwater Flow in the Timpoweap Canyon of Southwestern, Utah
130   [MS thesis]: Provo, Brigham Young University, 2005.

---

## Author Response (AR2)

**GEOSCIENCES**

September 23, 2020

*Solid Earth*

Dear Dr. Eichhubl,

Thank you for the additional comments on our revised manuscript. We are pleased to submit for your consideration our newly revised manuscript "Spatiotemporal history of fault-fluid interaction in the Hurricane fault, western U.S." by Jace M. Koger and Dennis L. Newell.

Below we detail how we addressed your questions and comments.

Best regards,

Dennis Newell

Corresponding Author

4505 Old Main Hill
Logan, UT 84322-4505
P-435-797-1273
F-435-797-1588
geo@usu.edu

[Figure]

Hi Dennis,

Thank you for your detailed response letter and your revisions. I have a few comments and questions on your revised ms and would appreciate if you could clarify, where appropriate, in the ms:

I am confused about the paragraph starting on line 585 of your tracked ms. "' the computed trends cannot explain the observed data"--you mean any single fractionation process in isolation, but to some extent the three types of fractionation processes would potentially occur together, wouldn't they? Are they mutually exclusive? I don't think they are. If that is correct, couldn't they account for the spread of data in Fig 4b without invoking non-equilibrium effects?

You seem to dismiss open-system conditions, but mass-balance considerations would make a closed system conditions difficult to justify. That would be a strong claim. For isolated veins in a tight rock sure, but where you have a distinct meteoric signal I would consider this as evidence for open conditions.

We agree that this discussion was a bit unclear and misleading, and the section has been updated. We have updated this to emphasize that the Rayleigh processes associated with progressive degassing and calcite mineralization cannot on their own explain the range of values observed and the trends, especially from a single starting fluid. We acknowledge that these processes are likely operable, however, and will impart a secondary impact on the isotopic values. In fact, these processes operating on a mixed fluid is probably the best way to account for the data variability.

Additionally, we agree that a closed system in not appropriate here. We have updated the sections to be clear that the fluid mixing and these Rayleigh processes are all occurring in an open system in the fault zone. Attaching "open system" to a Rayleigh process is redundant and causes confusion.

Ln 599: phenomenon is singular, either phenomena or a major phenomenon. Why not processes?

We have corrected the tense in this situation.

Ln 602: Two-phase trapping would be indicative of a rapid $CO_2$ degassing (i.e. $CO_2$ bubbles in the water) but this does not rule out slow $CO_2$ degassing at the water table. I think that would be expected in most moving meteoric systems (i.e. formation of speleothems in caves).

This was a suggestion from one of the reviewers that the lack of vapor dominated fluid inclusions supports the idea that rapid $CO_2$ degassing is not occurring. I think what he was alluding to is that you might expect to see a population of vapor dominated inclusions along with liquid dominated, similar to a boiling scenario. I do agree that slower degassing can form two-phase inclusions, or possibly the vapor ($CO_2$) and water were miscible at the time of trapping and cooling/exhumation has nucleated the second phase. We have updated this section for clarity.

Lastly, because you use SMOW for calcite d18O I get confused in some places in the manuscript if you are referring to calcite d18O or water d18O (first paragraph of section 5.2). My preference is to use PDB for calcite and SMOW for water, but if you use SMOW for calcite please make sure it is clear if you talk about calcite or (calculated) water values.

We have updated the methods, results, and discussion to make this clear. All of the isotopic

measurements from calcite are reported as VPDB, but where we discuss the calculated source fluids, we use VSMOW for the $\delta^{18}O$. Figure 4 was also a source of confusion because it mixes carbonate data with paleofluid trends. We make it clear that the carbonate oxygen data are reported on the VSMOW scale for the purposes of presenting the paleofluid trends on the same axes.

[revised manuscript text omitted]